# Molecular insights into the gating mechanisms of voltage-gated calcium channel CaV2.3

Yiwei Gao [1,2,3,11], Shuai Xu [4,11], Xiaoli Cui[2,3,5,11], Hao Xu[1,6], Yunlong Qiu[1,2,3], Yiqing Wei[1,2,3], Yanli Dong [1,2,3], Boling Zhu[7], Chao Peng[4], Shiqi Liu[4], Xuejun Cai Zhang [1,2,3], Jianyuan Sun[2,3,8,9], Zhuo Huang [4,10,12] ✉ & Yan Zhao [1,2,3,12] ✉

High-voltage-activated R-type CaV2.3 channel plays pivotal roles in many physiological activities and is implicated in epilepsy, convulsions, and other neurodevelopmental impairments. Here, we determine the high-resolution cryo-electron microscopy (cryo-EM) structure of human CaV2.3 in complex with the α2δ1 and β1 subunits. The VSD$_{II}$ is stabilized in the resting state. Electrophysiological experiments elucidate that the VSD$_{II}$ is not required for channel activation, whereas the other VSDs are essential for channel opening. The intracellular gate is blocked by the W-helix. A pre-W-helix adjacent to the W-helix can significantly regulate closed-state inactivation (CSI) by modulating the association and dissociation of the W-helix with the gate. Electrostatic interactions formed between the negatively charged domain on S6$_{II}$, which is exclusively conserved in the CaV2 family, and nearby regions at the alpha-interacting domain (AID) and S4-S5$_{II}$ helix are identified. Further functional analyses indicate that these interactions are critical for the open-state inactivation (OSI) of CaV2 channels.

Voltage-gated calcium (CaV) channels mediate calcium influx to cells in response to changes in membrane potential[1–3]. Their cellular roles have been emphasized for decades in a variety of studies and include hormone secretion[4,5], neurotransmitter release[6,7], and muscle contraction[8,9]. CaV channel members are categorized into the CaV1, CaV2, and CaV3 subfamilies based on sequence identity or alternatively classified into T-, L-, P/Q-, N-, and R-types according to their pharmacological and biophysical profiles[1]. The so-called pharmacoresistant (R-type) CaV2.3 is widely expressed in the brain and enriched in the hippocampus, cerebral cortex, amygdala, and corpus striatum[10–12]. Electrophysiological investigations revealed that currents mediated by CaV2.3 are resistant to common CaV blockers or gating modifiers such as nifedipine, nimodipine, ω-Aga-IVA, etc[13]. CaV2.3 channels exhibit cumulative inactivation in response to brief and repetitive depolarizations, a process known as preferential closed-state inactivation (CSI)[14]. Furthermore, CaV2.3 is involved in a broad spectrum of neuronal activities[10,12,15]. Previous studies have reported that CaV2.3 participates in multiple physiological processes in the central nervous

[1]National Laboratory of Biomacromolecules, CAS Center for Excellence in Biomacromolecules, Institute of Biophysics, Chinese Academy of Sciences, Beijing, China. [2]State Key Laboratory of Brain and Cognitive Science, Institute of Biophysics, Chinese Academy of Sciences, 15 Datun Road, Beijing, China. [3]College of Life Sciences, University of Chinese Academy of Sciences, Beijing, China. [4]State Key Laboratory of Natural and Biomimetic Drugs, Department of Molecular and Cellular Pharmacology, School of Pharmaceutical Sciences, Peking University Health Science Center, Beijing, China. [5]Chinese Institute for Brain Research, Beijing, China. [6]Division of Life Sciences and Medicine, University of Science and Technology of China, Hefei, China. [7]Center for Biological Imaging, Institute of Biophysics, Chinese Academy of Sciences, Beijing, China. [8]Sino-Danish College, University of Chinese Academy of Sciences, Beijing, China. [9]The Brain Cognition and Brain Disease Institute, Shenzhen Institute of Advanced Technology, Chinese Academy of Sciences (CAS), Shenzhen-Hong Kong Institute of Brain Science-Shenzhen Fundamental Research Institutions, Shenzhen, China. [10]IDG/McGovern Institute for Brain Research, Peking University, Beijing, China. [11]These authors contributed equally: Yiwei Gao, Shuai Xu, Xiaoli Cui. [12]These authors jointly supervised this work: Zhuo Huang, Yan Zhao. ✉e-mail: huangz@hsc.pku.edu.cn; zhaoy@ibp.ac.cn

system, such as inducing long-term potentiation (LTP) and post-tetanic potentiation in mossy fiber synapses[16], modulating the burst firing mode of action potentials[17,18], and regulating synaptic strength in hippocampal CA1 pyramidal neurons[19]. In recent years, increasing evidence has revealed that dysfunction of $Ca_V2.3$ is linked to epilepsy[20,21], convulsions[22,23], and neurodevelopmental impairments[24], suggesting that $Ca_V2.3$ is a pivotal player in the pathogenesis of a series of neurological disorders.

The molecular basis of $Ca_V$ channels has been investigated extensively over the past several decades, including structural studies of L-type $Ca_V1.1$[25,26], N-type $Ca_V2.2$[27,28], and T-type $Ca_V3.1$[29] and $Ca_V3.3$[30] in the apo form or distinct modulator-bound states. These structures provide substantial insights into the architecture, subunit assembly, and modulator actions of the $Ca_V$ channels. However, the gating mechanism of $Ca_V$ channels is still far from fully understood. For instance, in the $Ca_V2.2$ structure, the $VSD_{II}$ is trapped in a resting state by a $PIP_2$ molecule at a membrane potential of ~0 mV[27,28]. The functional roles of the $VSD_{II}$ trapped in the resting state by $PIP_2$ remain unknown. A considerable number of pathogenic mutations have been identified in the VSDs of neuronal $Ca_V2$ channels, demonstrating that VSD dysfunctions contribute to the genesis of spinocerebellar ataxia (SCA), episodic ataxia (EA), and familial hemiplegic migraine (FHM)[31]. Moreover, the $Ca_V2.2$ and $Ca_V2.3$ channels inactivate preferentially from the intermediate closed state along the activation pathway, which is important in controlling the short-term dynamics of synaptic efficacy[14,32]. In our previous study, we elucidated that residue W768 on the W-helix located within the DII-III linker serves as a key determinant of the CSI of the $Ca_V2.2$ channel[28]. However, $Ca_V2.3$ is characterized by a more prominent preferential CSI than $Ca_V2.2$[14]. It is also interesting to explore the modulation mechanism of CSI in $Ca_V2.3$. Furthermore, the high-voltage-activated (HVA) $Ca_V1$ and $Ca_V2$ channels harbor a conserved α-helix connecting Domain I and Domain II (alpha-interaction domain, or AID). Previous studies have indicated that AID might contribute to the open-state inactivation (OSI) of the HVA $Ca_V$ channels[33,34]. However, the inactivation properties of $Ca_V1$ and $Ca_V2$ channels are dramatically different[33]. Mechanistic insight into the inactivation processes of the HVA $Ca_V$ channels will help us to fully uncover the physiological role of $Ca_V$ channels and facilitate the development of therapeutic solutions for $Ca_V$-related diseases.

In this study, we expressed and purified human $Ca_V2.3$ in complex with auxiliary subunits α2δ1 and β1 and unveiled the high-resolution structure of this protein complex. Further mutagenesis and electrophysiological experiments were performed. Our results provide insights into the pharmacological resistance properties of $Ca_V2.3$, the asynchronous functional roles of the VSDs, the mechanism by which the pre-W-helix regulates the CSI, and the OSI process modulated by the negatively charged domain on $S6_{II}$ ($S6_{II}^{NCD}$).

## Results and discussion
### Architecture of the $Ca_V2.3$ complex
To gain structural insights into the $Ca_V2.3$ complex, we expressed and purified full-length wild-type human $Ca_V2.3$ α1E subunit (CACNA1E), α2δ1 (CACNA2D1) and β1 (CACB1) using a HEK 293-F expression system. The $Ca_V2.3$-α2δ1-β1 complex was solubilized using n-Dodecyl-β-D-maltoside (DDM) and purified using a strep-actin affinity column, followed by further purification by size-exclusion chromatography (SEC) in a running buffer containing glycol-diosgenin (GDN) to remove protein aggregates (Supplementary Fig. 1a, see Method section for details). The peak fractions were subsequently collected and concentrated for cryo-EM sample preparation (Supplementary Fig. 1b). A total of 2096 micrographs were collected. Data processing of the dataset gave rise to a 3.1-Å cryo-EM map, which is rich in high-resolution structural features, including densities for sidechains, lipid molecules, and glycosylations (Fig. 1, Supplementary Fig. 2, and Supplementary Table 1).

The $Ca_V2.3$ complex exhibited a conventional shape that resembles that of the $Ca_V2.2$ and $Ca_V1.1$ complexes (Fig. 1). The complex is composed of the transmembrane α1E subunit, extracellular α2δ1 subunit, and intracellular β1 subunit (Fig. 1a, b). The α1E subunit is a pseudo-tetrameric pore-forming subunit and can be divided into Domain I (DI) to IV (DIV). Each domain of the α1E subunit is composed of 6 transmembrane helices (S1–S6), comprising the voltage-sensing domain (VSD) (S1–S4) and the pore domain (S5–S6). The P1 and P2 helices located between the S5 and S6 helices formed the selectivity filter (Fig. 1c). Similar to other $Ca_V$ channels, $Ca_V2.3$ harbors four extracellular loops (ECLs) that are also positioned between S5 and P1, as well as between P2 and S6 helices in the pore domain (Fig. 1a, c). The $ECL_I$ and $ECL_{II}$ are critical for the association between the α1 subunit and the α2δ1 subunit (Fig. 1a, b). The S6 helices from the four domains converge on the cytoplasmic side and form the intracellular gate of the channel. In our structure, the intracellular gate was determined in its closed state, in line with the observations from other voltage-gated channels (Fig. 1c). Moreover, the closed gate of $Ca_V2.3$ is further stabilized by the W-helix from the DII-DIII linker, which is consistent with a previous study on $Ca_V2.2$ and indicates that $Ca_V2.3$ also adopts the CSI mechanism[28].

Most $Ca_V$ channels serve as pharmaceutical targets of a variety of small-molecule drugs or peptide toxins[13]. However, previous studies indicate that $Ca_V2.3$ is resistant to many $Ca_V$ modulators, such as nimodipine (L-type), omega-Aga-IVA (P/Q-type), and omega-CTx-GVIA (N-type)[13]. To clarify the structural basis underlying the pharmacoresistance of $Ca_V2.3$, we compared the structures of $Ca_V2.3$ with $Ca_V1.1$ or $Ca_V2.2$ in their ligand-bound states (Fig. 1d, e). In the structure of nifedipine-bound $Ca_V1.1$, the nifedipine molecule was located within the DIII-DIV fenestration and stabilized by surrounding residues. However, two critical residues are substituted in $Ca_V2.3$, namely, Y1296 and F1708. The bulky sidechains of these two residues in $Ca_V2.3$ occupy the DIII-DIV fenestration site and thus hinder the binding of nifedipine (Fig. 1d). Meanwhile, a previous study reported that the Q1010 of $Ca_V1.2$ is important for sensitivity to dihydropyridine (DHP) and the Q1010M mutant had a decreased sensitivity to DHP molecules[35]. The equivalent position in $Ca_V2.3$ is occupied by M1300, thus also contributing to the pharmacoresistance of $Ca_V2.3$ to DHP molecules. Structural comparison of $Ca_V2.3$ and ziconotide-bound $Ca_V2.2$ demonstrated that the $ECL_I$ loop of $Ca_V2.3$ adopts a different conformation, and residues D263 and P264 are placed close to the central axis, giving rise to clashes between the ziconotide and the $ECL_I$ of $Ca_V2.3$ (Fig. 1e). Moreover, other residues on P-loops and ECLs that are involved in ziconotide binding are also not conserved in $Ca_V2.3$ (Supplementary Fig. 3), rendering $Ca_V2.3$ insensitive to the ziconotide.

Gain-of-function mutations and polymorphisms of $Ca_V2.3$ channels have already been implicated in the pathological process of developmental and epileptic encephalopathy. Thirteen pathogenic mutations have been identified in $Ca_V2.3$[24]. Our high-resolution structure provides a structural template to map all of these pathogenic mutations, which are distributed throughout the complex structure (Fig. 1f). Eight of thirteen mutations are located around the intracellular gate, such as I603L, F698S, and I701V, and the result in a hyperpolarizing shift in the half-activation voltage[24] (Fig. 1f).

### Functional heterogeneity of the VSDs
The voltage-dependent gating characteristics of voltage-gated channels are conferred by their VSDs. The VSDs of $Ca_V$ are conserved helix bundles consisting of S1, S2, S3, and S4 helices (Supplementary Fig. 4a). S4 was found to be a positively charged $3_{10}$ helix, harboring approximately five or six arginines or lysines as gating charges lining one side of the helix at intervals of three residues (Supplementary Fig. 4a). The positively charged S4 helices move vertically toward the intracellular or extracellular side of the cell in response to the hyperpolarization or depolarization of the membrane potential. The

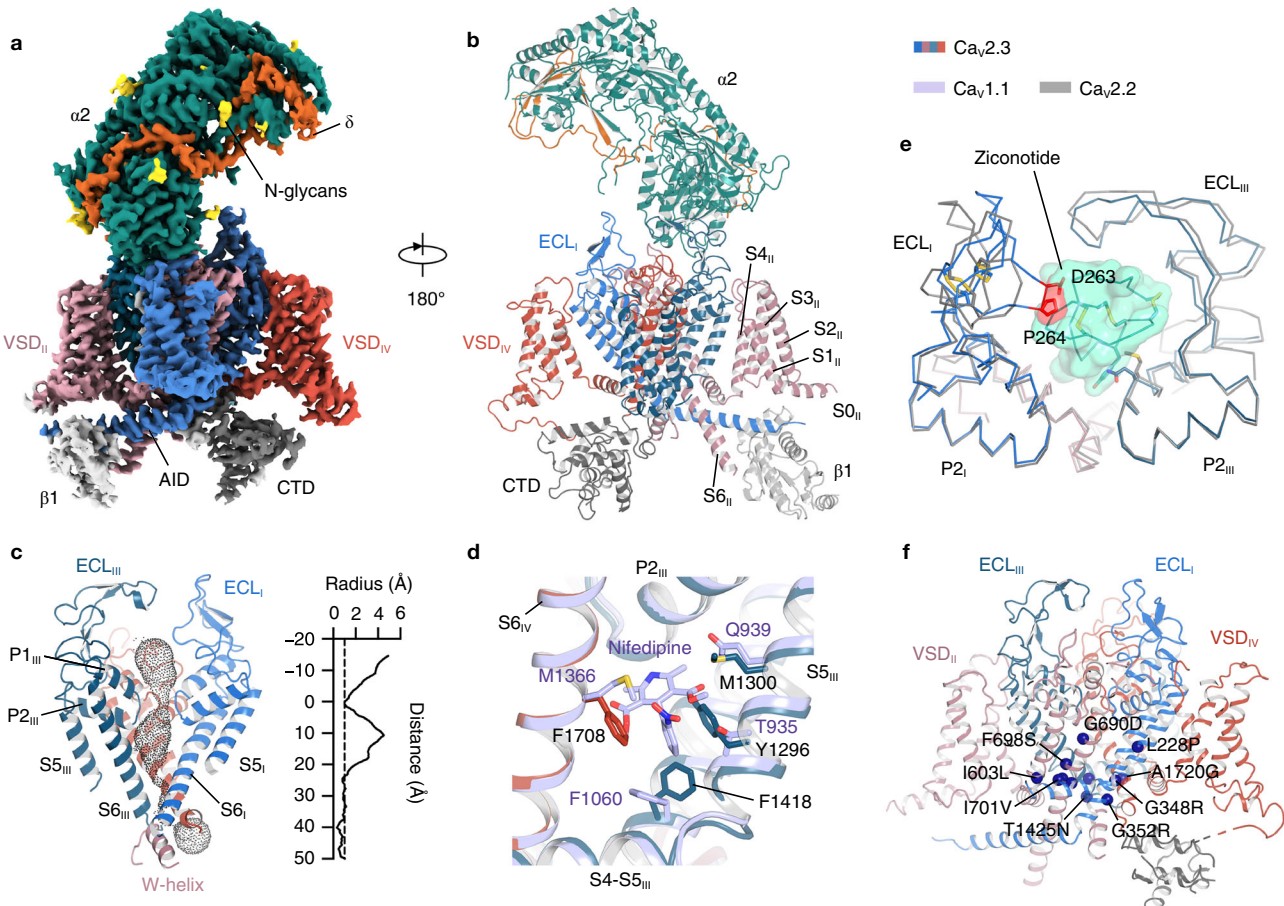

**Fig. 1 | Architecture of the Ca$_V$2.3-α2δ1−β1 complex. a, b** Cryo-EM map (**a**) and model (**b**) of the Ca$_V$2.3-α2δ1−β1 complex. The Ca$_V$2.3 α1E pore-forming subunit are colored pink, red, blue, and deep cyan. The C-terminal domain (CTD) of Ca$_V$2.3 are colored gray. The auxiliary subunits α2, δ, αvδ β1 are colored green, orange, and white, respectively. **c** The ion-conducting pathway and pore profiling of the Ca$_V$2.3. The ion-conducting pathway is viewed in parallel to the membrane and shown in black dots. The radius of the pore is calculated using the HOLE program. The vertical dashed line marks the 1.0-Å pore radius, which represents the ionic radius of calcium. **d** Superimposition of the DIII-IV fenestration between Ca$_V$2.3 and Ca$_V$1.1 complex. The nifedipine molecule

bound to the Ca$_V$1.1 complex is shown as sticks. Residues stabilizing the nifedipine molecule in Ca$_V$1.1, and residues that might form steric clashes in Ca$_V$2.3 are shown as sticks. **e** Superimposition between the 'P-loops' (P1 and P2) and extracellular loops (ECLs) of the Ca$_V$2.3 and the ziconotide-bound Ca$_V$2.2. The Ca$_V$2.3 and Ca$_V$2.2 are shown as ribbon, overlaid with the ziconotide shown as transparent green surfaces. Potential steric clash between the ECL$_I$ of Ca$_V$2.3 and the ziconotide is highlighted in red. **f** Pathogenic mutations of the Ca$_V$2.3, shown in blue spheres. The four domains of Ca$_V$2.3 are colored pink, red, blue, and deep cyan, respectively. Ca$_V$1.1 is colored purple and Ca$_V$2.2 is colored gray.

conformational change of VSD is coupled to the pore domain by a short amphipathic helix S4-S5, which connects the S4 helix of VSD to the S5 helix from the pore domain, thus regulating the transition of the intracellular gate between the open and closed states. Although the four VSDs of Ca$_V$ channels are considerably similar in terms of sequence and overall structure, they contribute differentially to the opening of pore[36].

Superimposition of the structures of Ca$_V$2.3 and Ca$_V$2.2 revealed that they are comparable overall (r.m.s.d. = 1.46 Å for 2222 Cα atom pairs). The pore domain was fairly superimposable between Ca$_V$2.3 and Ca$_V$2.2, including the S5 and S6 helices and extracellular loops (ECLs) I, III, and III (Supplementary Fig. 4b). VSD$_I$, VSD$_{III}$, and VSD$_{IV}$ in the activated state and VSD$_{II}$ in the resting state were also determined in both Ca$_V$2.3 and Ca$_V$2.2 (Supplementary Fig. 4a, b). However, a structural discrepancy was visualized at the ECL$_{IV}$ between the two structures. The ECL$_{IV}$ of Ca$_V$2.3 extends from the pore domain and lies above the S1-S2$_{III}$ linker, whereas the ECL$_{IV}$ of Ca$_V$2.2 is much shorter and wraps around the pore domain before touching the extracellular side of VSD$_{III}$ (Fig. 2a). Four residues on ECL$_{IV}$ of Ca$_V$2.3, namely, P1680, D1681, T1682, and T1683, are involved in the interactions with the residues on the S1-S2$_{III}$ linker, especially V1176, L1177, T1178, and N1179, which

consequently stitches the VSD to the pore domain at the extracellular side (Fig. 2a). To explore the functional roles of this interaction, we substituted [1680]PDTT[1683] on ECL$_{IV}$ with four glycines (Ca$_V$2.3[4G]) to disrupt the contacts between ECL$_{IV}$ and S1-S2$_{III}$ loop. Electrophysiological studies indicated that the voltage dependency of the activation curve of Ca$_V$2.3[4G] displayed a ~5 mV positive shift ($P < 0.0001$, two-tailed unpaired $t$ test) compared to that of wild-type Ca$_V$2.3 (Fig. 2b and Supplementary Fig. 5). We thus speculate that the interactions between ECL$_{IV}$ and the S1-S2$_{III}$ loop may stabilize the VSD$_{III}$ in a certain conformation relative to the pore domain that requires less electrical energy to activate the channel, reminiscent of the cholesterol regulation on Ca$_V$ channels, potentially by stabilizing interactions between the extracellular end of the S1-S2 helix hairpin and the pore domain[37,38].

The VSD$_{II}$s of Ca$_V$2.3 and Ca$_V$2.2 were determined in the resting (S4 down) state, while most other VSDs in the voltage-gated channels, such as Ca$_V$1.1, Ca$_V$3.1, and Na$_V$s, were determined in the activated (S4 up) state (Supplementary Fig. 4c, d). This finding suggests that VSD$_{II}$ plays a unique role in the gating mechanism among Ca$_V$2 members. Structural analyses of Ca$_V$2.2 have suggested that the VSD$_{II}$ is trapped in the resting state by a PIP$_2$ molecule[27,28]. The cryo-EM map of Ca$_V$2.3 is of high quality around S4-S5$_{II}$ (Supplementary Fig. 2e), and we

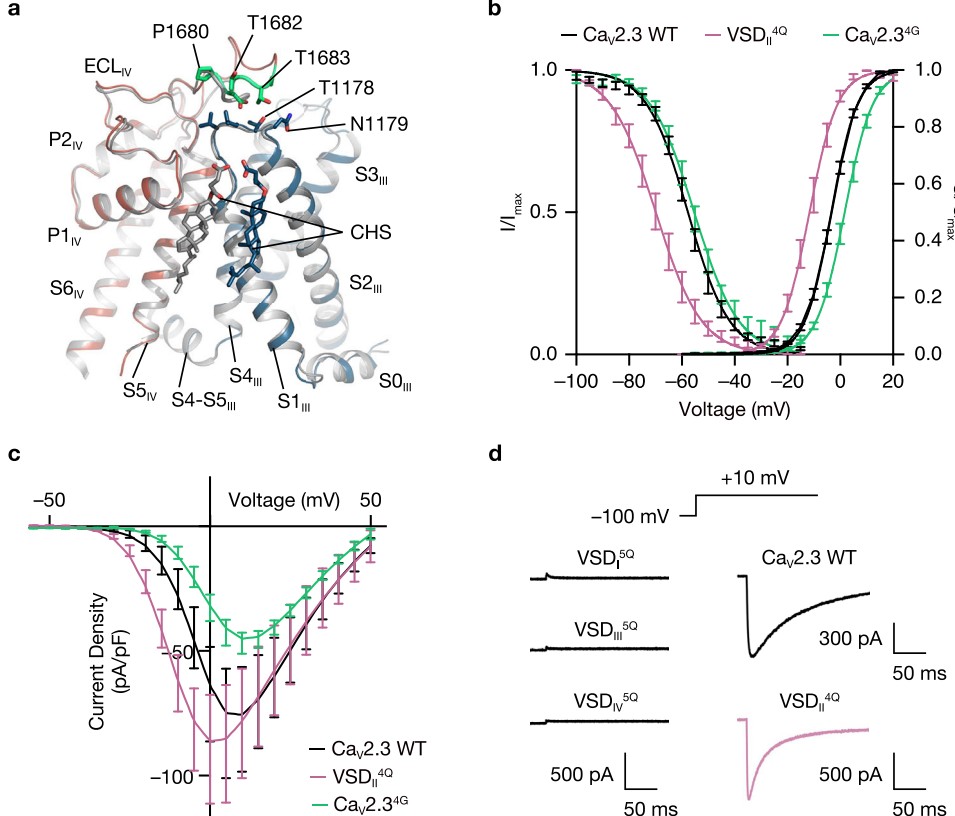

**Fig. 2 | Voltage-sensing domains of the Ca$_V$2.3 complex. a** Superimposition of the VSDs (DIII) and the pore domains (DIV) of the Ca$_V$2.3 (red and blue) and Ca$_V$2.2 (gray) demonstrating the interaction between the ECL$_{IV}$ and the VSD$_{III}$. Residues on ECL$_{IV}$ in close contact with VSD$_{III}$ ($^{1680}$PDTT$^{1683}$) are shown in sticks and colored green. Cholesteryl hemisuccinate (CHS) molecules are shown as sticks and labeled. **b** Steady-state activation and inactivation curves of the wild-type (WT) Ca$_V$2.3 (black), VSD$_{II}^{4Q}$ (pink), and Ca$_V$2.3$^{4G}$ (green). To examine the voltage dependence of activation, HEK 293-T cells expressing the Ca$_V$2.3 complex were tested by 200-ms depolarizing pulses between −60 and 50 mV from a holding potential of −100 mV,

in 10-mV increments. To determine inactivation curves, cells were stepped from a holding potential of −100 mV to pre-pulse potentials between −100 and −15 mV in 5-mV increments for 10 s. Activation curve, Ca$_V$2.3 WT, $n = 14$; VSD$_{II}^{4Q}$, $n = 7$; Ca$_V$2.3$^{4G}$, $n = 11$. Steady-state inactivation curve, Ca$_V$2.3 WT, $n = 9$; VSD$_{II}^{4Q}$, $n = 7$; Ca$_V$2.3$^{4G}$, $n = 9$. **c** Current density of the wild-type Ca$_V$2.3 and VSD$_{II}^{4Q}$. Ca$_V$2.3 WT, $n = 8$; VSD$_{II}^{4Q}$, $n = 8$; Ca$_V$2.3$^{4G}$, $n = 7$. **d** Voltage-clamp protocol and representative traces of the Ca$_V$2.3 WT, VSD$_{I}^{5Q}$, VSD$_{II}^{4Q}$, VSD$_{III}^{5Q}$, and VSD$_{IV}^{5Q}$, elicited by a 200-ms test pulse at +10 mV. Data are presented as mean ± SEM. $n$ biological independent cells.

identified a single strip-shaped density that does not look like a PIP$_2$ molecule. However, another recent structural investigation of Ca$_V$2.3 channel showed that PIP$_2$ could bind to this site but is not responsible for the resting state of VSD$_{II}$[39]. To shed light on the functional roles of VSD$_{II}$ in the gating mechanism of Ca$_V$2.3, we constructed a gating charge neutralization mutant on the VSD$_{II}$ (VSD$_{II}^{4Q}$, R572Q/R575Q/R578Q/K581Q). Interestingly, the neutralization mutation of the VSD$_{II}$ (VSD$_{II}^{4Q}$) exhibits -9-mV left shifts in the voltage dependency of both activation and steady-state inactivation compared to the wild-type (Fig. 2b and Supplementary Fig. 5a, b). Consequently, the current density-voltage curve of the VSD$_{II}^{4Q}$ mutant was also left-shifted (Fig. 2c). Moreover, we tested whether the VSD$_{II}^{4Q}$ mutant has a distinct CSI profile. The cumulative inactivation of this mutant in response to action potential (AP) trains is markedly enhanced (Supplementary Fig. 5c, e). These results suggested that the conformation of VSD$_{II}$ influences the gating of Ca$_V$2.3. However, its OSI kinetics remain unaltered (Supplementary Fig. 5d). In contrast, the gating charge-neutralized mutation in the VSD$_{I}$, VSD$_{III}$, and VSD$_{IV}$ resulted in failure to mediate inward current (Fig. 2d), in line with previous results showing that the VSD$_{I}$, VSD$_{III}$, and VSD$_{IV}$ are important for gating of the closely-related Ca$_V$2.2[40–42], while the VSD$_{II}$ is not necessary for channel activation by sensing the depolarization of membrane potential; instead, the VSD$_{II}$ is crucial to modulating channel properties, such as CSI and voltage dependency of channel activation and inactivation.

## Molecular mechanism of closed-state inactivation

Preferential closed-state inactivation is a featured kinetic characteristic of neuronal Ca$_V$ channels[14,28]. During the state-transition pathway in the activation of Ca$_V$ channels, CSI occurs preferentially in a specific pre-open closed state and in a voltage-dependent manner. CSI can be visualized by the cumulative inactivation in response to action potential (AP) trains, as reported in previous studies, which demonstrated that the peak current triggered by each AP shrank sequentially, suggesting that a substantial amount of the channels turn inactivated after the repolarization of an AP. CSI plays an important role in the orchestrated modulation mechanism of Ca$_V$ channels and is of vital importance for the precise regulation of physiological processes such as neurotransmitter release and synapse plasticity. CSI is detected in all neuronal Ca$_V$2 channels at distinct levels[14]. The R-type Ca$_V$2.3 displayed a more prominent CSI than the N-type Ca$_V$2.2, and both showed far more prominent CSI than the P/Q-type Ca$_V$2.1[14]. Previous structural investigations on the N-type Ca$_V$2.2 channel have revealed that a conserved W-helix is the structural determinant of CSI, and W768 from the W-helix on the DII-DIII linker functions as a lid blocking the pore and stabilizes the intracellular gate in its closed states by hydrophobic interactions. However, the molecular mechanisms underlying the CSI of Ca$_V$s are still not fully understood, as the conserved W-helix is unable to explain the disparity of CSI mechanisms among the Ca$_V$2.1, Ca$_V$2.2, and Ca$_V$2.3 channels.

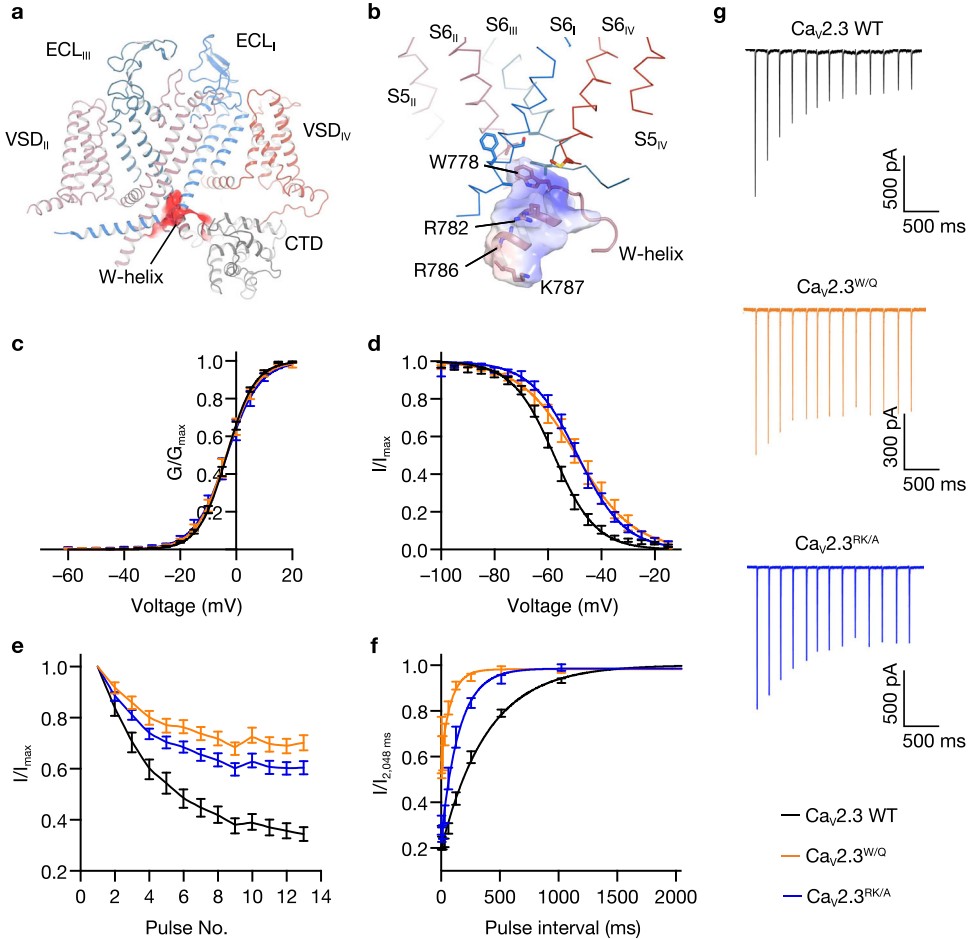

**Fig. 3 | Closed-state inactivation mediated by W-helix. a** Binding pocket of the W-helix. The W-helix is accommodated in a negatively charged pocket on the intracellular side of Cav2.3. The Cav2.3 α1E subunit is shown in the cartoon, and the negatively charged pocket accommodating W-helix is shown as red surface. **b** Zoomed-in view of the W-helix, which is stabilized in the intracellular gate at a closed state. The W-helix is shown in cartoon and overlaid with an electrostatic surface. W778 and other residues involved in the hydrophobic or charge interactions are shown in sticks. **c** Activation curves of the wild-type (WT) Cav2.3 and the mutants. Cav2.3 WT, $n = 14$; Δ$w$-helix, $n = 7$; Cav2.3RK/A, $n = 7$. **d** Steady-state inactivation curves of Cav2.3 WT and the mutants. Cav2.3 WT, $n = 9$; Δ$w$-helix, $n = 8$; Cav2.3RK/A, $n = 10$. **e** Inactivation ratio quantified using the current density (I) elicited by each spike of AP trains divided by the maximum current ($I_{Peak}$) elicited by the

first spike. Cav2.3 WT, $n = 13$; Cav2.3 W/Q, $n = 15$; Δ$w$-helix, $n = 9$; Cav2.3RK/A, $n = 14$. **f** Recovery rate from the CSI, quantified by the inactivation ratio between the two peak currents (I and $I_{2,048ms}$) obtained in the two-pulse protocol. HEK 293-T cells were held at −40 mV for 1500 ms and were then stepped to −100 mV for a series of time intervals (4–2,048 ms) before a +10 mV test pulse (35 ms). Cav2.3 WT, $n = 6$; Cav2.3 W/Q, $n = 6$; Δ$w$-helix, $n = 6$; Cav2.3RK/A, $n = 7$. **g** Representative current responses to the AP trains. The AP trains used to stimulate the HEK 293-T cells were recorded from a mouse hippocampal CA1 pyramidal neuron after current injection in whole-cell current-clamp mode. See Supplementary Figure 5 for voltage-clamp protocols and Method section for literature reference. Cav2.3 WT, black; Cav2.3 W/Q, orange; Cav2.3RK/A, blue. Data are presented as mean ± SEM. $n$ biological independent cells.

The W-helix determined in Cav2.2 is also conserved and well-resolved in Cav2.3 (Fig. 3a, b). The W-helix in Cav2.3 (772RHHMSVWEQRTSQLRKH788) is a positively charged short helix and positioned underneath the intracellular gate, with W778 inserting into the gate and forming extensive interactions with residues from surrounding gating helices (Fig. 3a, b). These structural observations are consistent with the W-helix of Cav2.2. First, we designed the Cav2.3 W/Q (W778Q) to disrupt interactions between the W-helix and intracellular gate (Fig. 3c–g and Supplementary Fig. 6). It turns out that the Cav2.3 W/Q exhibited a −8-mV positive shift on the steady-state inactivation curve (Fig. 3d) and an alleviated cumulative inactivation in response to AP trains (Fig. 3e, g) without affecting the voltage dependence of channel activation (Fig. 3c), consistent with the observations in Cav2.2[28], suggesting that the W778 is important for CSI process of Cav2.3 channel. Moreover, we speculate that the positively charged residues on the W-helix could putatively respond to the membrane potential change and may be important for the initiation of the CSI process. To evaluate our speculations, we constructed the Cav2.3RK/A mutant by substituting

the R781, R786, and K787 with alanine (R781A/R786A/K787A). The activation curve of the Cav2.3RK/A mutant remains unaltered (Fig. 3c). However, the CSI of Cav2.3RK/A mutant was significantly suppressed, exhibiting a ~9 mV right shift on the steady-state inactivation curve (Fig. 3d), an alleviated cumulative inactivation to AP trains (Fig. 3e, g), and an accelerated recovery rate from CSI (Fig. 3f and Supplementary Fig. 6b). These alterations on the CSI profile suggested that the positively charged R781, R786, and K787 are critical for the CSI mechanism of Cav2.3.

Intriguingly, sequence alignment among the neuronal Cavs revealed that a peptide segment (pre-W-helix) that resembled the W-helix is located adjacent to the W-helix of Cav2.3 (Fig. 4a). The sequence of pre-W-helix (753RHHMSMWEPRSSHLRER769) is nearly identical to that of the W-helix (~65% identity), including the conserved tryptophan plug (W759) and positively charged residues (R762, R767, and R769) (Fig. 4a). However, we noticed that a non-conserved proline (P761) was in the middle of the pre-W-helix, which may undermine the stability of both the pre-W-helix itself and its interactions with the gate.

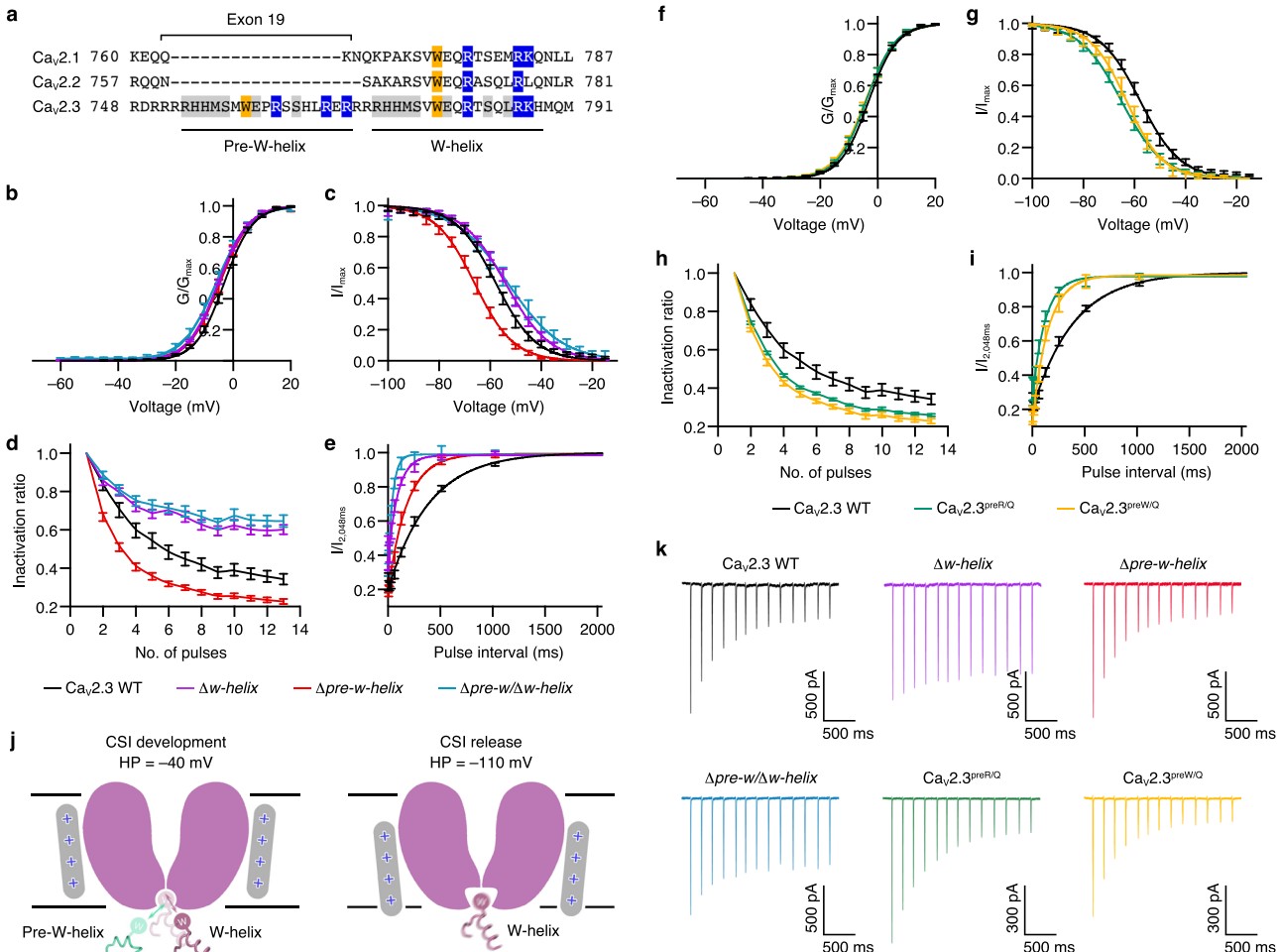

**Fig. 4 | CSI regulation by pre-W-helix. a** Sequence alignment of the DII-III linker among the Ca$_V$2 members. Residues on the pre-W-helix and W-helix are underlined and labeled, respectively. Exon 19 on the genomic DNA of Ca$_V$2.3 encoding the pre-W-helix is also labeled. **b, g** Activation curves of the wild-type Ca$_V$2.3 and the mutants. Ca$_V$2.3 WT, $n=14$; $\Delta w$-helix, $n=7$; $\Delta pre$-$w$-helix, $n=7$; $\Delta pre$-$w$/$\Delta w$-helix, $n=7$; Ca$_V$2.3$^{preR/Q}$, $n=7$; Ca$_V$2.3$^{preW/Q}$, $n=8$. **c, h** Steady-state inactivation curves of wild-type Ca$_V$2.3 and the mutants. Ca$_V$2.3 WT, $n=9$; $\Delta w$-helix, $n=8$; $\Delta pre$-$w$-helix, $n=7$; $\Delta pre$-$w$/$\Delta w$-helix, $n=7$; Ca$_V$2.3$^{preR/Q}$, $n=8$; Ca$_V$2.3$^{preW/Q}$, $n=6$. **d, i** Inactivation ratio quantified using I/I$_{Peak}$ elicited by the AP trains. Ca$_V$2.3 WT, $n=13$; $\Delta w$-helix, $n=9$; $\Delta pre$-$w$-helix, $n=13$; $\Delta pre$-$w$/$\Delta w$-helix, $n=14$; Ca$_V$2.3$^{preR/Q}$, $n=9$; Ca$_V$2.3$^{preW/Q}$, $n=13$. **e, j** Recovery rate from the CSI, quantified using the previously described two-pulse protocol. Ca$_V$2.3 WT, $n=6$; $\Delta w$-helix, $n=6$; $\Delta pre$-$w$-helix, $n=7$; $\Delta pre$-$w$/$\Delta w$-helix, $n=6$; Ca$_V$2.3$^{preR/Q}$, $n=8$; Ca$_V$2.3$^{preW/Q}$, $n=7$. **f** Proposed model of the putative

regulatory role of the pre-W-helix in the CSI of Ca$_V$2.3. Membrane planes are indicated using gray lines. Pore domains are depicted as cartoon and colored purple. Positively charged S4 helices are shown as gray rounded bars. Pre-W-helix and the W-helix are shown as cartoon. In the development process of CSI (left), the intracellular gate might have a high affinity to W-helix. As a consequence, W-helix and pre-W-helix competitively bind to the gate. In the release process of CSI (right), the gate might adopt an alternative conformation and the W-helix is likely to disassociate from the gate. **k** Representative current responses stimulated by AP trains. See Supplementary Figure 5 for voltage-clamp protocols and Method section for literature reference. Ca$_V$2.3 WT, black; $\Delta w$-helix, purple; $\Delta pre$-$w$-helix, red; $\Delta pre$-$w$/$\Delta w$-helix, green-cyan; Ca$_V$2.3$^{preR/Q}$, green; Ca$_V$2.3$^{preW/Q}$, yellow. Data are presented as mean ± SEM. $n$ biological independent cells.

Moreover, in our cryo-EM map of the Ca$_V$2.3 complex, the helical density beneath the gate perfectly fits the atomic model of the W-helix, enabling us to unambiguously determine that the W-helix, instead of the pre-W-helix, exists in our structure (Supplementary Fig. 2e). Considering that the pre-W-helix is located immediately before the W-helix and that they share high sequence identity, we speculate that the pre-W-helix participates in the CSI event of Ca$_V$2.3. To evaluate the contribution of the pre-W-helix to the CSI of Ca$_V$2.3, we constructed two mutants by deleting the W-helix ($\Delta w$-helix) and pre-W-helix ($\Delta pre$-$w$-helix) (Fig. 4b–e, 4k, and Supplementary Fig. 6). The $\Delta w$-helix exhibited a ~4-mV positive shift on the steady-state inactivation curve (Fig. 4c) and alleviated cumulative inactivation in response to AP trains (Fig. 4d, k) without affecting the voltage dependence of channel activation (Fig. 4b), indicating that the W-helix plays pivotal roles in the CSI of Ca$_V$2.3. However, compared with the CSI of Ca$_V$2.2, which was almost abolished by deleting the W-helix, a substantial portion of the CSI in

the $\Delta w$-helix mutant remained unaltered (Fig. 4d, k), suggesting that the CSI modulation of Ca$_V$2.3 is distinct from that of Ca$_V$2.2 and that other elements may also contribute to the CSI of Ca$_V$2.3. We also used a two-pulse protocol to assess the recovery rate from CSI, i.e., the release process of CSI (Supplementary Fig. 6a). Consistent with the electrophysiological results above, an accelerated recovery rate from CSI was observed in the $\Delta w$-helix compared to the WT (Fig. 4e and Supplementary Fig. 6b). Strikingly, a negative-shift of ~8 mV was detected on the inactivation curve of the $\Delta pre$-$w$-helix mutant (Fig. 4c), and its cumulative inactivation to AP trains was surprisingly enhanced (Fig. 4d, k), demonstrating that the development process of CSI in the $\Delta pre$-$w$-helix was significantly boosted. Considering that the pre-W and W helices are close to each other and share high sequence identity, we speculate that the pre-W-helix may serve as a competitive negative regulator to interfere with the binding of the intracellular gate to the W-helix (Fig. 4f). In the absence of the pre-W-helix, the CSI is

consequently enhanced (Fig. 4c, d, k). Paradoxically, compared with the WT, the recovery rate from CSI of the $\Delta pre$-$w$-$helix$ mutant was substantially accelerated (Fig. 4e and Supplementary Fig. 6b). This suggests that once CSI occurs, the pre-W-helix appears to stabilize the W-helix for interaction with the gate during membrane potential repolarization, thereby slowing the recovery of Ca$_V$2.3 from the CSI. We also designed a mutant by deleting both the pre-W-helix and the W-helix ($\Delta pre$-$w$/ $\Delta w$-$helix$). This mutant displayed a ~5-mV positive shift on the inactivation curve (Fig. 4c), reduced cumulative inactivation to AP trains (Fig. 4d, k), and an accelerated recovery rate from CSI (Fig. 4e and Supplementary Fig. 6b). These effects of this double deletion construct are essentially identical to those of the $\Delta w$-$helix$, demonstrating that the modulatory role of the pre-W-helix on CSI is largely dependent on the W-helix, probably by regulating the binding or dissociation of the W-helix with the intracellular gate. To further investigate the regulatory mechanism on the CSI of Ca$_V$2.3, we constructed two mutants by neutralizing the arginines (Ca$_V$2.3$^{preR/Q}$, R753Q/R762Q/R767Q/R769Q) or substituting the tryptophan on the pre-W-helix with glutamine (Ca$_V$2.3$^{preW/Q}$, W759Q) (Fig. 4g–k and Supplementary Fig. 6). Impressively, the mutants Ca$_V$2.3$^{preR/Q}$ and Ca$_V$2.3$^{preW/Q}$ exhibit similar gating kinetics and voltage dependence of channel activation and inactivation, but significantly different from that of the WT Ca$_V$2.3 channel (Fig. 4g–k, Supplementary Fig. 5a, b and Supplementary Fig. 6b). In particular, they displayed a ~7-mV negative shift on inactivation curve (Fig. 4h), enhanced cumulative inactivation to AP trains (Fig. 4i, k), and an accelerated recovery rate from CSI (Fig. 4j and Supplementary Fig. 6b), highly identical to the CSI profile of the $\Delta pre$-$w$-$helix$. The kinetic characteristics of these mutants indicated that the positively charged residues and W759 are important for the regulatory effect of the pre-W-helix. Our data also show that the development and release of CSI are two independent processes. In particular, the $\Delta pre$-$w$-$helix$, Ca$_V$2.3$^{preR/Q}$, and Ca$_V$2.3$^{preW/Q}$ exhibited enhanced cumulative inactivation during AP trains (Fig. 4d, i, k) but a faster recovery rate from CSI (Fig. 4e, j and Supplementary Fig. 6b) at a membrane potential of −100 mV, suggesting that the channel may have distinct conformational states along the activation pathway. The W-helix may bind preferentially to the channel in the intermediate closed state(s) near the open state, thus leading to the largest inactivation before the channel opens (Fig. 4f). At strongly hyperpolarized potentials (-100 mV), the channel is probably stabilized in a different closed state, exhibiting a lower affinity to the W-helix, causing the W-helix to detach from the gate and the channel to recover from the CSI (Fig. 4f). Taken together, the pre-W-helix in Ca$_V$2.3 plays significant roles in regulating the association or dissociation of the W-helix with the intracellular gate and thus exerts a regulatory effect on the development and release of the CSI.

Interestingly, on the genomic DNA of Ca$_V$2.3, the pre-W-helix is encoded by exon 19 (residues R748–R769) (Fig. 4a). Early studies have reported that alternative splicing events might occur at this site, as exon 19 is spliced out in the mature mRNA encoding Ca$_V$2.3e, an isoform of Ca$_V$2.3 that displays decreased Ca$^{2+}$ sensitivity in its calcium-dependent modulation mechanisms[43]. Gene expression profiling has revealed that Ca$_V$2.3e is enriched in endocrine tissues, including the kidney and pancreas, and the majority of Ca$_V$2.3 in the brain contains the pre-W-helix[44]. This implies that the kinetic variability of Ca$_V$2.3 mediated by the pre-W-helix is an important regulatory mechanism to fine-tune the properties of the Ca$_V$2.3 channel to adapt to the distinct physiological needs of neuronal and endocrinal excitable cells[43].

## Modulation of open-state inactivation

Upon the opening of the pore, Ca$_V$ channels undergo an inactivation process called open-state inactivation (OSI), which describes the mechanism by which the intracellular gate shifts swiftly from the open state into the inactivation state[33,45]. This process is an intrinsic mechanism that precisely regulates calcium influx into cells during depolarization. Neuronal (P/Q-, N- and R-type) Ca$_V$2 channels bear a stronger OSI and mediate a rapidly inactivated current, while cardiac (L-type) Ca$_V$ channels display a much weaker OSI, mediating a long-lasting current[46]. Dysfunction of OSI, i.e., the gain-of-function of neuronal Ca$_V$s, is linked to a series of neurological disorders, including trigeminal neuralgia[47], myoclonus-dystonia-like syndrome[48], and epileptic encephalopathies[21,24]. Although the structures of L-type Ca$_V$1.1[25] and N-type Ca$_V$2.2[27,28] complexes have been elucidated at high resolution, the structural basis for their distinct OSI properties remains elusive. Intriguingly, when comparing the EM density maps of the Ca$_V$1.1, Ca$_V$2.2, and Ca$_V$2.3 complexes, we observed that AID displayed a blurred density in Ca$_V$1.1 but was clearly resolved in Ca$_V$2.2 and Ca$_V$2.3 (Supplementary Fig. 7). Previous studies suggested that the AID helix is able to regulate the OSI in Ca$_V$ channels[33,34].

In the structure of Ca$_V$2.3, the S6$_{II}$ helix is much longer than that of Ca$_V$1.1 and extends into the cytosol. Interestingly, we identified a negatively charged domain $^{715}$DEQEEEE$^{721}$ in the intracellular juxtamembrane region of the S6$_{II}$ helix (S6$_{II}$$^{NCD}$) (Fig. 5a, b). Taking a closer look at the structure, this negatively charged region forms electrostatic interactions with the positively charged R590 on S4-S5$_{II}$, as well as R371 and R378 on the AID (Fig. 5b). Strikingly, the S6$_{II}$$^{NCD}$ is conserved among P/Q-, N- and R-type channels (Fig. 5c). The equivalent segment of L-type Ca$_V$ channels contains fewer negatively charged residues, interspersed with some positively charged residues, indicating that the electrostatic interactions between S6$_{II}$$^{NCD}$ and AID are not present in the L-type Ca$_V$ channels, which is consistent with structural observations that the AID of Ca$_V$1.1 has high motility (Fig. 5c and Supplementary Fig. 7). Additionally, the positively charged R590 and R378 are conserved only in Ca$_V$2 channels. R378 is further reverted to a negatively charged glutamate in the Ca$_V$1 subfamily, reflecting the coevolutionary linkages within this interaction site (Fig. 5c). We speculate that the charge interactions centered on S6$_{II}$$^{NCD}$ is critical to regulating the OSI of Ca$_V$2 channels.

To validate our hypothesis, we mutated the key residues to disrupt electrostatic interactions cross-linking the S6$_{II}$, S4-S5, and AID helices, including replacing $^{715}$DEQEEEE$^{721}$ with $^{715}$NNQNNNN$^{721}$ (Ca$_V$2.3$^{NQ}$), R590Q, R378Q/E, and R371Q/E (Fig. 5d and Supplementary Fig. 8). We employed Ba$^{2+}$ as the charge carrier in our whole-cell patch-clamp analysis to exclude the effects of calcium-dependent inactivation (CDI) of the Ca$_V$2.3 channels. The Ca$_V$2.3$^{NQ}$ mutant mediates a current that decays much more slowly than the wild-type Ca$_V$2.3 during a 200-ms test pulse (Supplementary Fig. 8a, b), suggesting that OSI is remarkably decreased. To quantify the OSI of the Ca$_V$2.3 mutants, we employed the R200 value as an indicator, which is calculated by the mean current density at the end of the 200-ms test pulse divided by the peak amplitude (Fig. 5d and Supplementary Fig. 8c). Specifically, the mean value of R200 increased from $0.17 \pm 0.01$ in the wild-type Ca$_V$2.3 to $0.31 \pm 0.03$ in Ca$_V$2.3$^{NQ}$ under the test pulse holding at 10 mV (Fig. 5d and Supplementary Fig. 8c). Moreover, R590E and R590Q also exhibited remarkably suppressed OSI, with increased R200 values of $0.35 \pm 0.04$ and $0.27 \pm 0.03$, respectively (10-mV test pulse) (Fig. 5d and Supplementary Fig. 8c), suggesting that the R590-S6$_{II}$$^{NCD}$ interaction plays an essential role in the development of OSI in Ca$_V$2 channels. In contrast, mutants on the AID side show complicated effects on OSI kinetics (Fig. 5d and Supplementary Fig. 8c). In particular, R378Q displayed an enhanced OSI, with a decreased R200 value of $0.11 \pm 0.02$ (10-mV test pulse). R371E, which is located in the adjacent region of R378, exhibited an enhanced OSI as well, displaying an R200 value of $0.09 \pm 0.01$ (10-mV test pulses). Nevertheless, OSI of the R371Q and R378E mutants do not show a significant difference with that of the WT Ca$_V$2.3 channel (Fig. 5d and Supplementary Fig. 8c). Moreover, the time course of channel inactivation could be well fitted by a single exponential. Conclusions drawn using time constants are nearly identical to those using R200 values (Supplementary Fig. 8d), further supporting

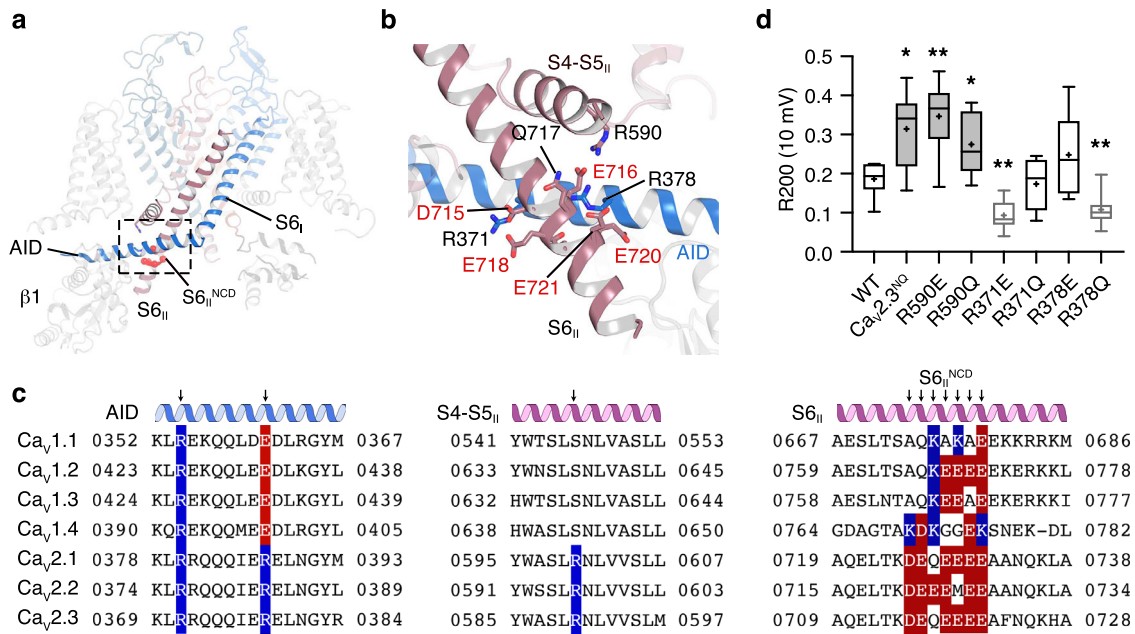

**Fig. 5 | Modulation of open-state inactivation. a** Interaction of the AID between the negatively charged domains on S6$_{II}$ (S6$_{II}^{NCD}$) and S4-S5$_{II}$. The Ca$_V$2.3 α1E subunit is shown as cartoon. The S6$_{II}^{NCD}$ is overlaid as red spheres. **b** Zoomed-in view of the side-chain interactions among S6$_{II}^{NCD}$, AID, and S4-S5$_{II}$. The S6$_{II}^{NCD}$ ($^{715}$DEQEEEE$^{721}$) is shown as sticks. Residues involved in charge interactions on AID and S4-S5$_{II}$ are also shown as sticks. Negatively charged residues within the S6$_{II}^{NCD}$ are highlighted using red labels. **c** Sequence alignment of the interaction sites among Ca$_V$1 and Ca$_V$2 members. Secondary structures are labeled over the sequences. Mutation sites are indicated by arrows. Residues with positively- and negatively charged sidechains are highlighted by blue and red, respectively. **d** Ratio of open-state inactivation (R200) at 10-mV test pulses, measured using the mean current at the end of the 200-ms test pulse divided by the peak amplitude. WT, $n = 6$; Ca$_V$2.3$^{NQ}$, $n = 10$; R590E, $n = 6$; R590Q, $n = 10$; R378E, $n = 6$; R378Q, $n = 9$; R371E, $n = 6$; R371Q, $n = 9$. Data are plotted as box plots. The box encompasses the interquartile range (25th–75th percentile). Whiskers illustrates the minima and maxima of the values. Mean values and medians are indicated using plus signs and dashes, respectively. Significances were determined using two-sided, unpaired $t$ test. $P$ values, Ca$_V$2.3 WT vs. mutants; 0.01 (Ca$_V$2.3$^{NQ}$), 0.005 (R590E), 0.02 (R590Q), 0.004 (R371E), and 0.004 (R378Q). $n$ biological independent cells.

that S6$_{II}^{NCD}$, R371 (AID), R378 (AID), and R590 (S4-S5$_{II}$) play important roles in OSI modulation. However, the interaction between the AID and S6$_{II}^{NCD}$ could go beyond the current structures of Ca$_V$ channels and is worth investigating in future studies.

## Methods

### Expression and protein purification of the human Ca$_V$2.3 complex

Full-length Ca$_V$2.3 α1E (CACNA1E), α2δ1 (CACNA2D1), and β1 (CACB1) were amplified from a human cDNA library and subcloned into pEG BacMam vectors. To detect the expression and assembly levels of the Ca$_V$2.3 complex, a superfolder GFP, an mCherry, and an mKalama tag were fused to the C-terminal, N-terminal, and N-terminal regions of the Ca$_V$2.3 α1, α2δ1, and β1 subunits, respectively. Twin-Strep tags were tandemly inserted into the C-terminal and N-terminal of the α1 and α2δ1 subunits, respectively. Primers are provided in Supplementary Table 2. The Bac-to-Bac baculovirus system (Invitrogen, USA) was used to conduct protein expression in HEK 293-F cells (Thermofisher, 11625019) following the manufacturer's protocol. The bacmids were prepared using DH10Bac competent cells, and P1 viruses were generated from Sf9 cells (Thermofisher, 10902096) after bacmid transfection. P2 viruses (1%, v/v) were used to infect HEK 293-F cells supplemented with 1% (v/v) fetal bovine serum. The cells were cultured at 37 °C and 5% CO$_2$ for 12 h before the addition of 10 mM sodium butyrate to the medium. The cells were cultured at 30 °C and 5% CO$_2$ for another 48 h before harvest. No authentication was performed for the HEK 293-F or the Sf9 cell line. No Mycoplasma contamination was observed.

The cell pellets were resuspended at 4 °C using Buffer W containing 20 mM HEPES pH 7.5, 150 mM NaCl, 5 mM β-mercaptoethanol (β-ME), 2 μg/mL aprotinin, 1.4 μg/mL leupeptin, and 0.5 μg/mL

pepstatin A (MedChemExpress, USA) by a Dounce homogenizer, followed by centrifugation at 110,000 × g for 1 h to collect the membrane. The membrane was resuspended again using Buffer W and solubilized by the addition of 1% (w/v) n-dodecyl-β-D-maltoside (DDM) (Anatrace, USA), 0.15% (w/v) cholesteryl hemisuccinate (CHS) (Anatrace, USA), 2 mM adenosine triphosphate (ATP) and 5 mM MgCl$_2$ on a rotating mixer at 4 °C for 2 h. The addition of ATP and MgCl$_2$ is to remove associated heat shock proteins. The insoluble debris of cells was removed by another centrifugation at 110,000 × g for 1 h. The supernatant was passed through a 0.22 μm filter (Millipore, USA) before being loaded into 6 mL Streptactin Beads 4FF (Smart-Lifesciences, China). The resin was washed using 6 column volumes of Buffer W1 (20 mM HEPES pH 7.5, 150 mM NaCl, 5 mM β-ME, 0.03% (w/v) glycol-diosgenin (GDN) (Anatrace, USA), 2 mM ATP, and 5 mM MgCl$_2$). The purified Ca$_V$2.3 complex was eluted using 15 mL elution buffer (20 mM HEPES pH 7.5, 150 mM NaCl, 5 mM β-ME, 0.03% (w/v) GDN, and 5 mM d-Desthiobiotin (Sigma-Aldrich, USA)) and concentrated to 1 mL using a 100 kDa MWCO Amicon (Millipore, USA). The concentrated protein sample was subjected to further size-exclusion chromatography (SEC) by a Superose 6 Increase 10/300 GL gel filtration column (GE Healthcare, USA) using a flow rate of 0.3 mL/min and a running buffer containing 20 mM HEPES pH 7.5, 150 mM NaCl, 5 mM β-ME, and 0.01% (w/v) GDN (Anatrace, USA). The monodispersed peak fraction within 12–13.5 mL was pooled and concentrated to 5 mg/mL before preparing the cryo-EM grids.

### Cryo-EM sample preparation and data collection

Holey carbon grids (Au R1.2/1.3 300 mesh) (Quantifoil Micro Tools, Germany) were glow-discharged using H$_2$ and O$_2$ for 60 s before being loaded with 2.5 μL purified Ca$_V$2.3 complex. The grids were automatically blotted for 4 s at 4 °C and 100% humidity and flash-frozen in

liquid ethane using a Vitrobot Mark IV (Thermo Fisher Scientific, USA). Cryo-EM data were collected using a 300 kV Titan Krios G2 (Thermo Fisher Scientific, USA) equipped with a K2 Summit direct electron detector (Gatan, USA) and a GIF Quantum LS energy filter (Gatan, USA). The dose rate was set to ~9.2 e⁻/(pixel*s), and the energy filter slit width was set to 20 eV. A total exposure time of 6.72 s was dose-fractioned into 32 frames. A nominal magnification of ×130,000 was used, resulting in a calibrated super-resolution pixel size of 0.52 Å on images. SerialEM[49] was used to automatically acquire the movie stacks. The nominal defocus range was set from −1.2 μm to −2.2 μm.

## Cryo-EM data processing

Motion correction was performed on 2096 movie stacks using MotionCor2[50] with 5 × 5 patches, generating dose-weighted micrographs. The parameters of the contrast transfer function (CTF) were estimated using Gctf[51]. Particles were initially picked using the blob picker in cryoSPARC[52], followed by 2D classifications to produce 2D templates and ab initio reconstruction to generate an initial reference map. Another round of particle picking was conducted using Template Picker in cryoSPARC, generating a dataset of 787,518 particles that were used for further processing. All data processing steps were performed in RELION-3.1[53] unless otherwise specified. A round of multi-reference 3D classification was conducted against one good and 4 biased references, generating 5 classes. Class 5 (75.2%), which was calculated using the good reference, displayed a classical shape of Ca$_V$ complexes featuring a transmembrane subunit and two soluble subunits residing on both sides of the micelle. Particles from class 5 were re-extracted and subjected to another round of 3D classification, resulting in 6 classes. Classes 1, 4, and 5 (43.5%) displayed well-resolved structural features, including continuous transmembrane helices of the α1E subunit and secondary structures within the α2δ1 and β1 subunits. To improve the map quality, Bayesian polish and CTF refinement were then conducted. The following 3D auto refinement generated a 3.1-Å map. The particle dataset was then imported back to cryoSPARC, where the final map was generated by Non-uniform (NU) refinement, which was reported at 3.1 Å according to the golden-standard *Fourier* shell correlation (GSFSC) criterion.

## Model building

The cryo-EM map of Ca$_V$2.3 was reported at near-atomic resolution, which enabled us to reliably build and refine the model. The structure of the Ca$_V$2.2-α2δ1-β1 complex (PDB ID: 7VFS)[28] was selected as the starting model because of the high sequence identity and was docked into the map of Ca$_V$2.3 complexes using UCSF Chimera[54]. Sidechains of the a1 subunit were manually mutated according to the sequence alignment between Ca$_V$2.3 and Ca$_V$2.2 and adjusted according to the EM density using *Coot*[55]. Sidechains of α2δ1 were also manually adjusted according to the EM density. β1 was initially fit into the EM maps as a rigid body and manually refined against a low-resolution map of the Ca$_V$2.3 complex in *Coot* due to local structural heterogeneity. The manually adjusted models were then automatically refined against the cryo-EM maps using the integrated Real Space Refinement program within the PHENIX software package[56]. Model stereochemistry was also evaluated using the Comprehensive validation (cryo-EM) tool in PHENIX.

All the figures were prepared using Open-Source PyMOL (Schrödinger, USA), UCSF Chimera[54], or UCSF ChimeraX[57].

## Whole-cell voltage-clamp recordings of Ca$_V$2.3 channels in HEK 293-T cells

HEK 293-T cells were cultured with Dulbecco's modified Eagle's medium (DMEM) (Gibco, USA) supplemented with 15% (v/v) fetal bovine serum (FBS) (PAN-Biotech, Germany) at 37 °C with 5% CO$_2$. The cells were grown in culture dishes ($d$ = 3.5 cm) (Thermo Fisher Scientific,

USA) for 24 h and then transiently transfected with 2 μg of control or mutant plasmid expressing the human R-type Ca$_V$2.3 calcium channel complex (Ca$_V$2.3 α1E, β1, α2δ1) using 1.2 μg of Lipofectamine 2000 Reagent (Thermo Fisher Scientific, USA). Patch-clamp experiments were performed 12 to 24 h post-transfection at room temperature (21-25 °C) as described previously. Briefly, cells were placed on a glass chamber containing 105 mM NaCl, 10 mM BaCl$_2$, 10 mM HEPES, 10 mM D-glucose, 30 mM TEA-Cl, 1 mM MgCl$_2$, and 5 mM CsCl (pH = 7.3 with NaOH and an osmolarity of ~310 mosmol⁻¹). Whole-cell voltage-clamp recordings were made from isolated, GFP-positive cells using 1.5-2.5 MΩ fire-polished pipettes (Sutter Instrument, USA) filled with standard internal solution containing 135 mM K-gluconate, 10 mM HEPES, 5 mM EGTA, 2 mM MgCl$_2$, 5 mM NaCl, and 4 mM Mg-ATP (pH = 7.2 with CsOH and osmolarity of ~295 mosmol⁻¹). Whole-cell currents were recorded using an EPC-10 amplifier (HEKA Electronik, Germany) at a 20 kHz sample rate and were low-pass filtered at 5 kHz. The series resistance was 2-4.5 MΩ and was compensated 80-90%. The data were acquired by the PatchMaster program (HEKA Electronik, Germany).

To obtain activation curves of Ca$_V$2.3 channels, cells were held at −100 mV, and then a series of 200-ms voltage steps from −60 mV to +50 mV in 5-mV increments were applied. The steady-state inactivation properties of Ca$_V$2.3 channels were assessed with 10-s holding voltages ranging from −100 mV to −15 mV (5-mV increments) followed by a 135-ms test pulse at +10 mV. To assess the time-dependent recovery from CSI, cells were depolarized to −40 mV (pre-pulse) for 1500 ms to allow Ca$_V$2.3 channels to enter CSI, and recovery hyperpolarization steps to −100 mV were applied for the indicated period (4 ms−2048 ms), followed by a 35-ms test pulse at +10 mV. To assess the cumulative inactivation of Ca$_V$2.3 channels in response to AP trains, the cells were held at −100 mV, and then the physiologically relevant AP trains was applied. The AP trains used to stimulate the HEK 293-T cells were recorded from a mouse hippocampal CA1 pyramidal neuron after current injection in the whole-cell current-clamp mode[58]. The spike pattern contained 13 action potentials in 2 s (mean frequency = 6.5 Hz). The percentage inactivation of Ca$_V$2.3 channels was calculated from the first spike eliciting maximal current to the other spikes in the AP trains. To analyze the extent of OSI, the ratio of remaining currents at 200 ms post-depolarization and the peak currents was calculated.

## Electrophysiological data analysis

All data are reported as the mean ± SEM. Data analyses were performed using Origin 2019b (OriginLab, USA) and Prism 9 (GraphPad, USA).

Steady-state activation curves were generated using a Boltzmann Eq. (1).

$$\frac{G}{G_{max}} = \frac{1}{1 + \exp(V - V_{0.5})/k} \tag{1}$$

where $G$ is the conductance, calculated by $G = I/(V - V_{rev})$, where $I$ is the current at the test potential and $V_{rev}$ is the reversal potential; $G_{max}$ is the maximal conductance of the Ca$_V$2.3 channel during the test pulse; $V$ is the test potential; $V_{0.5}$ is the half-maximal activation potential; and $k$ is the slope factor.

Steady-state activation and inactivation curves were generated using a Boltzmann Eq. (2).

$$\frac{I}{I_{max}} = \frac{1}{1 + \exp(V - V_{0.5})/k} \tag{2}$$

where $I$ is the current at the indicated test pulse; $I_{max}$ is the maximal current of Ca$_V$2.3 activation during the test pulse; $V$ is the test potential; $V_{0.5}$ is the half-maximal inactivation potential; and k is the slope factor.

Recovery curves from CSI were calculated from the results of 7−9 independent experiments where a series of recovery traces from inactivation time points were acquired. The data were fit using a single exponential of the following Eq. (3).

$$\frac{I}{I_{\max}} = (y_0 - 1){}^*\exp\left(-\frac{t}{\tau}\right) + 1 \qquad (3)$$

where $I$ is the current at the indicated intervals; $I_{\max}$ is the current at 2048 ms; $y_0$ is the remaining current at −40 mV for 1500 ms; $t$ is the indicated hyperpolarization time; and τ is the time constant of recovery from CSI.

Statistical significance ($p < 0.05$) was determined using unpaired Student's $t$ tests or one-way ANOVA with Tukey's post hoc test.

### Reporting summary

Further information on research design is available in the Nature Portfolio Reporting Summary linked to this article.

## Data availability

The data that support this study are available from the corresponding authors upon reasonable request. The cryo-EM density map of the Ca$_V$2.3-α2δ1-β1 complex has been deposited in the Electron Microscopy Data Bank (EMDB) under the accession code EMD-33285. The coordinate for the Ca$_V$2.3 complex have been deposited in the Protein Data Bank (PDB) under the PDB ID 7XLQ. The starting model used to build Ca$_V$2.3-α2δ1-β1 is available in the PDB under the PDB ID 7VFS (Ca$_V$2.2-α2δ1-β1). DNA sequences of human Ca$_V$2.3 α1E (CACNA1E isoform 1), human α2δ1 (CACNA2D1), and human β1 (CACB1) are available in the Universal Protein Resource (UniProt) databases under accession codes Q15878-1 [https://www.uniprot.org/uniprotkb/Q15878/entry], P54289, and Q02641, respectively. Source data including an uncropped scan of the gel image and values of all electrophysiological experiment graphs are provided as Source Data Files. Source data are provided in this paper.

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

## Acknowledgements

We thank Xiaojun Huang, Xujing Li, Lihong Chen, and other staff members at the Center for Biological Imaging (CBI), Core Facilities for Protein Science at the Institute of Biophysics, Chinese Academy of Science for their support in cryo-EM data collection. We thank Yan Wu for his research assistance. This work is funded by the National Key Research and Development Program of China (Grant No. 2021YFA1301501 to Y.Z.), the Chinese Academy of Sciences Strategic Priority Research Program (Grant No. XDB37030304 to Y.Z.; Grant No. XDB37030301 to X.C.Z.), the National Natural Science Foundation of China (Grant No. 92157102 to Y.Z.; Grant No. 31971134 to X.C.Z.; Grant No. 81371432 to Z.H., and U20A6005 to J.S.), Chinese National Programs for Brain Science and Brain-like Intelligence Technology (Grant No. 2022ZD0205800 to Y.Z.; Grant No. 2021ZD0202102 to Z.H.), and National Science and Technology Major Project (2021ZD0202501 to J.S.).

## Author contributions

Y.Z. conceived the project and supervised the research. Y.G. and Y.Q. carried out molecular cloning experiments. Y.G. expressed and purified protein samples. Y.D. prepared samples for cryo-EM study. Y.G. and B.Z. carried out cryo-EM data collection. Y.G. processed the cryo-EM data. Y.G. built and refined the atomic model. Y.Z., Y.G., X.C.Z., and Y.W. analyzed the structure. Y.Z., Z.H., Y.G., S.X., and J.S. designed the electrophysiological experiments. S.X., X.C., H.X., C.P., and S.L. conducted the whole-cell voltage patch-clamp analysis. Y.G. wrote the original draft of the manuscript and prepared the figures. Y.Z., Y.G., and X.C.Z. edited the manuscript with input from all authors.

## Competing interests

All authors declare no competing interests.
