## [Peer Review File · Nature Communications]

Molecular insights into the gating mechanisms of voltage-gated calcium channel CaV2.3Reviewers' Comments:

Reviewer #1:

Remarks to the Author:

The manuscript by Y Gao et al "Molecular insights into the gating kinetics of voltage-gated calcium channel Cav2.3" is devoted to structural and functional study of Cav2.3 channel. The topic of study is actual; the work has excellent novelty and executed using state-of-the-art methodology. To my opinion, the manuscript could be accepted for publication in Nature Communications after minor revision.

There are several major and lot of minor concerns, which should be addressed by the authors before the acceptance.

Major points:

- 1) When describing obtained functional/mutagenesis results, the authors often use the term "kinetics of gating/inactivation". However, in most cases, they did not study the kinetics (the rate of processes), they simply measured some properties of the activation/inactivation processes, e.g. shift of the activation curve along the voltage axis. The obtained data give information about energy changes during transitions, but do not provide the information about the kinetics of gating process.
- 2) The manuscript contains very large number of typos and small mistakes, including reference to the figures, mutant naming etc.
- 3) Many figures are drawn using pale colors and too small for understanding. They should be either redrawn using more contrast palette or substantially increased in size.
- 4) Some important experimental details about sample preparation (e.g. detergent composition) are missing.
- 5) The data for W778 mutant is required.

Detailed description and minor points:

- 1) Introduction section mentions that there are several types of Cav channels L-, N-, T-, etc. Almost all the channels are attributed to some type, but not the Cav2.3 itself. Please provide the type of this channel (R-) and briefly describe the major biophysical properties of this type of channels.
- 2) Introduction, line 77. Abbreviation CSI is used before its introduction at line 78.
- 3) Results and Discussion, lines 95-105. Please provide the detergent composition of the sample.
- 4) Figure S1. Please provide details of SEC (detergent and buffer composition, column, flow rate).
- 5) Figure S1 caption. β 1 instead of δ 1.
- 6) Figure S2 caption. "Heterologous refinement and non-uniform (NU) refinement in generated the final map, which was reported at 3.1 Å" something missing after "in".
- 7) Figure S2 caption. S6II instead of S4II.
- 8) Line 114, please mention the helices that surround ECLs. (e.g. ECLs located between S5, P1, P2, or S6 helices in pore domain).
- 9) Line 120, please repeat once more that W-helix belongs to D2-D3 linker.
- 10) Lines 131-134, the reference to figure 1e is missing.
- 11) Lines 139-141, incorrect figure reference. Should be 1f instead of 1d.
- 12) Line 141, the literature reference(s) supporting the statement about shift in the half-activation voltage is needed.
- 13) Figure 1, panel f is unclear. It should be redrawn using bright colors, or whole figure 1 should be resized.
- 14) Lines 167,170. Loop instead of loops.
- 15) Line 171. Incorrect use of term "gating kinetics". Provided data do not give any information about kinetics.
- 16) Line 170. Too complicated speculation about influence of ECL on coupling of the VSD to the gate. Probably situation is simpler. ECL stabilize the VSD in some conformation, that change it probability to switch in the activated state (requiring less electric force). Please describe this. The gate of the channel is too far from the ECL.
- 17) Line 174. The cholesterol/lipid molecules shown on the figure 2a is not described in the main text or figure caption.
- 18) Figure S3. Please show the W-helix and pre-W-helix (if it observable in your structure) on the

panels b-d.

- 19) Figure 2. The trace for the WT channel should be added to the panel d.
- 20) Caption to the figure 2. The "minus" signs are missed before 100 and 15 mV. (-100 and -15)
- 21) Line 185. Mutant instead of mutants.
- 22) Lines 192-193. The conclusion about the alternative conformation of VSD-4Q is obvious, but how this state differs from the resting state is unknown. I propose to change this conclusion to the "conformation of VSD-II influences gating of the channel".
- 23) Line 194. Kinetics of OSI on the figure S4a was not analyzed. Please either provide results of R200 analysis (like in Figure S7) or rephrase.
- 24) Lines 196-199. This statement represents major result of the corresponding section. Please consider variant to move it in the beginning of the section. The fact that VSD-2 influences (not controls) the voltage-gated activation should be mentioned. For the VSD-4Q mutant you see not only CSI changes, but voltage shift of the activation curve.
- 25) Figure S4. The caption describing panel c is unclear. Please provide more details for measuring AP trains in hippocampus and mention that the responses of mutated Cav2.3 channels were recorded in the HEK cells overexpressing the channels.
- 26) Figure S4. Data for 4G mutant should be added to the panel d.
- 27) Figure S4d. Please clarify which statistic test was used.
- 28) Lines 222-223 and Figure 3. The panel 3a is too small and not bright. Please, consider other colors or resize the panel. The full gray surface of alpha subunit is unnecessary.
- 29) Lines 226, 231. K787 should be instead of R787.
- 30) Line 227. Please mention, important observation, that activation of the channel is not changed by RK/A mutation
- 31) Lines 219-231. It is very interesting what happens upon mutation of W778 residue. This data is needed to support conclusions of the work.
- 32) Line 248. "kinetics of channel activation" was not measured.
- 33) Figures 3 and 4 contain similar data and identical curves for WT and -dW-helix channels. Please consider to join these figures in one. The source data for RK/A mutant should be shown, like in the panel 4k.
- 34) Figure 4. Caption to panel k is unclear, please see above. The reference to Figure S5 should be changed to the reference for Figures S4 and S5.
- 35) Figure 4. It is very inconvenient to show only one legend (with mutant/coloring encoding) for such a large multi-panel figure. Please consider variant to provide separate legends for panels b-e and g-j.
- 36) Figure S5. The panel a is redundant, it repeats the figure S4c (left). Please remove.
- 37) Line 279. Not only kinetics but other properties are similar for preW/Q and preR/Q mutants. Please mention it.
- 38) Line 286. Figure 4d and 4i instead of I and k.
- 39) Line 287. Figure 4e instead of Figure 4.
- 40) Line 335. Figure 5c instead of 5d.
- 41) Line 345. The sentence "S6IINCD plays central roles in connecting with both AID and the S4-S5 helix." is unsupported conclusion and not connected with surrounding text fragments.
- 42) Line 364. Please mention that R590 residue belongs to S4-S5 linker.
- 43) Figure 5d. Please clarify, which statistic test was used.
- 44) Lines 389-390. Please provide description, why ATP and MgCl₂ are needed for channel purification.
- 45) Lines 396-397. Please provide composition of elution buffer. (including detergent composition).
- 46) Lines 398-400. Please provide composition of buffers used for SEC. (including loading buffer and elution buffer).

Reviewer #2:

Remarks to the Author:

Yiwei Gao and collaborators present the first high-resolution (3.1 Angstrom) cryo-EM structure of the

human R-type voltage gated calcium channel CaV2.3. The structure of the pore forming subunit alpha1E is resolved in complex with the accessory subunits alpha2delta1 and beta1 providing the original information about the organization of this macromolecule. CaV2.3 pore is in the closed state stabilized by interactions with the PreW- and W- helices, suggesting a mechanisms for closed-state inactivation. The voltage sensors are in the active position with the exception of VSDII in which the S4 segment is held into its resting position by specific interactions with an unknown molecule.

By comparing CaV2.3 and Cav1.1 pore regions, the Authors identified two residues that pose steric hindrance to nifedipine binding, thus providing a reasonable structural basis for the DHP resistance of Cav2.3. Similarly, the structural basis for Cav2.3 resistance to ziconotide is revealed by comparison with a ziconotide-bound Cav2.2 structure.

Overall, the presented structural work is valuable and has the potential to advance the field: it contributes a high-resolution structure of a new voltage gated calcium channel, shedding light on key mechanisms of channels gating and pharmacoresistance.

However, I have several concerns about functional studies included in this manuscript: they should be addressed. I also find that some of the conclusions are not supported by specific data (e.g. lack of experimental results supporting the proposed role of the voltage sensing domains).

Some of the statements require qualification and/or references.

There are many mistakes and typos.

I have offered my comments and suggestions below.

Main Points:

1. The authors want to assign a role to the interaction between the ECL-IV and the VSDIII. Are G/Gmax curves for WT and 4G (Fig2 B) significantly different?

What are the evidences that CaV 3.2 VSDIII is energetically coupled to the pore?

The 5 mV shift in the GV curve as consequence of the 4G mutation in ECL-IV may be due to a destabilization of the open state and not a change in coupling of VSDIII, as suggested. The slope of Cav2.34G G/Gmax curve should decrease when a VSD that contributes to voltage dependent activation is partially uncoupled from the pore, as it is proposed for the 4G mutant. Instead, it looks like Cav2.34G produce a parallel shift of the G/Gmax curve (same slope as WT G/Gmax) (Fig. 2B). In addition, to fully establish a functional role for these interactions, mutations in S1-S2-III liker (V, L, T, N residues) should be tested: in principle, they should cause similar changes in channel voltage dependence.

2. Please, in Fig.2C, display the IV curve for Cav2.3-4G together with with WT and VSDII-4Q.

3. Conclusions regarding the role of VSDII and the 4Q mutant are vague and unclear.

Line 192: the Authors conclude that "the VSDII of VSDII4Q mutant may adopt an alternative conformation other than the resting state in the wild-type CaV2.3." This seems an important point of the study; however, this "alternative conformation" is not explained. Likewise, how this alternative conformation relates to the observed effect on inactivation and voltage dependence is not discussed.

4. Line 189: The Authors states that current density of the VSDII4Q mutant is "slightly increased" (Fig. 2C). Is this a statistically significant change? Why is it important to make this point? If the Authors think that this is due to the negative shift of $V_{1/2}$ of activation, perhaps they should say so, unless the mutant's surface expression is higher than WT.

5. Lines 347-348 regarding Open State Inactivation: the authors state that "CaV2.3NQ mutant mediates a current that decays much more slowly than the wild-type CaV2 suggesting that OSI is largely abolished". The Inactivation process is still very prominent in the NQ mutant thus claiming that OSI is "largely abolished" is not correct.

6. The neutralization of the charged residues in VSD I, III and IV did not elicit functional channels so no data are presented about the properties of channel with the impaired VSDs (except for VSDII). The Authors state that "These results indicate that the VSDI, VSDIII, and VSDIV play indispensable roles in voltage-gated pore opening". I do not think such a conclusion can be made from non-functional channels (Fig. 2D).

7. Line 74: The statement that Cav2.3 channels inactivate from an intermediate closed state needs a reference(s).

8. Line 115: "In addition, the alpha2delta1 subunit interacts with the S1-S2I loop, implying that alpha2delta1 modulates the gating kinetics of CaV2.3". Why does this interaction imply a modulation of gating kinetics? Please explain and provides specific reference(s).

9. The manuscript includes a full section entitled "Functional heterogeneity of the voltage-sensing domains". This is the title of a published paper (Pantazis et al. PNAS 2014) that first demonstrated "functional heterogeneity" of Cav channel voltage sensors: there is no reference to that (or related) studies.

10. Fig 4F is supposed to offer a model for the regulatory role of pre-W helix. As it is, the cartoon is not very helpful and does not explain how Pre-W and W helices work to tune CSI. The inhibitory (or "competitive negative regulator") role of Pre-w does not seem captured by the drawing.

11. Line 327: The Authors mention the identification of a negatively charge domain, 715 EEQEEEE 721 in the intracellular region of the S6II helix (Figure 5a-5b). In the Cav2.3 structure (Fig. 5), residue 715 is Asp; If this is correct, the domain is DEQEEEE not EEQEEEE. This is also in Figure 5 legend.

12. OSI-related mutants and Suppl. Fig 7. The representative current recordings for both R378Q and R378E clearly display faster inactivation then WT. Nevertheless, in the plot of OSI ratios (R200, Suppl. Figure 7) R378Q appears to be slower than WT.

Also, R200 values for R371E and R378E are practically identical in spite of the exceptional difference in their inactivation kinetics shown on the recordings.

There is an obvious discrepancy between some of the representative current families and the estimated extent of inactivation shown in the R200 plot. The simple R200 method does not seem adequate to describe the phenomena the Authors are attempting to study here. Perhaps, a more accurate estimate of the inactivation properties using single or, if necessary, double exponential fitting of the current decay, could provide more consistent results and facilitate their interpretation.

Other points

1. Suppl. Fig.5B: in Cav2.3preR/Q, the peak current amplitude after 2s recovery (rightmost peak) is smaller than the vertical arrow that represents (I am guessing) the peak inward current amplitude after a recovery time of 2048ms. Please explain or correct.

2. Suppl. Fig 1 b: Please indicate in the legend the expected MW of alpha1, alpha2delta and beta1

3. Line 284: "the development and release of CSI is asynchronous". Should be "are".

4. Line 284: "Interestingly, our data also suggest that the development and release of CSI is asynchronous". It is not clear what asynchronous means here. Can development and release of CSI be "synchronous"? Please explain why this is an interesting finding.

5. Line 316: "Although recent investigations have revealed the high-resolution cryo-EM structures of the CaV1.1 (L-type) and CaV2.2 (N-type) complexes, their imparity on OSI kinetics has not yet been explained". Not a clear sentence; please rephrase.
6. Line 324: "Previous studies suggested..." but only a single study is reported (30).
7. Line 334: "which is consistent with structural observations that the AID of CaV1.1 has high motility (Figure 5d and Supplementary Figure 6)." Figure 5d shows R200 ratios and does not seem relevant here.
8. Suppl. Fig 1 legend: "delta 1" should be "beta 1'.
9. Fig 4e: X axis is mislabeled: it should be "Time" not "Voltage".
10. Line 195: "In contrast, the gating charge-neutralized mutation in the VSDI, VSDII, and VSDIV resulted in failure to mediate inward current". Should be "VSDI, VSDIII, and VSDIV."
11. Figure 5 title. "Structural basis of the modulation on OSI mechanism". Should be "of OSI mechanism".

Reviewer #3:

Remarks to the Author:

Voltage-gated calcium channels mediate calcium ion flux across the membrane of excitable cells. Among the calcium channels, Cav2.3 plays an important role in neurotransmitter release and is associated with neuronal dysfunctions such as epilepsy. However, there was little available structural information of Cav2.3 channel and thus understanding of the molecular mechanism of this channel has been hindered. In this paper, Gao et al. provided structural insights on pharmacology, gating kinetics, and inactivation of Cav2.3. They have determined a high-resolution cryo-EM structure of human Cav2.3 in complex with its auxiliary units. The results provide the detailed mechanisms underlying the molecular function of Cav2.3 channel. The studies are straightforward, well-designed and the results are convincing and clearly described. The conclusions are supported by structural and functional approaches. But a few things need to be addressed before publication.

1. Line 99 - 101:

The authors already purified the complex and successfully determined the structure, therefore the statement for monodispersity and particle distribution in this part is unnecessary and misleading. We cannot say that particles are distributed homogeneously with this result. The SEC trace displays that multiple species are in the peak and the peak seems a sum of several peaks with some high backgrounds. The author's statement here is refuted by the evidence they provided, Supplementary figure 1. In the supplementary figure 1b the SDS-PAGE shows many bands, so it is unlikely that protein particles were distributed homogenously. In protein purification, it is not surprising that some minor impurities, aggregates and cleaved products are in a peak. And the minor impurities as well as some heterogeneous particles can be removed during EM data analysis. I believe that taking out this part does not harm the excellence of this study.

2. Supplementary Figure 1b:

Because of the multitude of protein bands, it is hard to know which bands are matched for Cav2.3 $\alpha 1E$, $\alpha 2\delta 1$, and $\beta 1$. Additional marks such as arrows or lines are necessary to help readers to understand the SDS-PAGE gel.

3. Line 122-134:

The Cav2.3 structure and comparing it with other Cav channels show structural basis underlying the pharmacoresistant property of Cav2.3. The Cav2.3 structure is of high quality and clearly shows why nifedipine and ziconotide cannot bind to Cav2.3. To further support this finding, relevant mutagenesis and functional studies testing nifedipine and ziconotide insensitivity are necessary. For example, mutations on D263 and P264 to make Cav2.3 sensitive to ziconotide will be of interest and can be supportive of their structural findings.

4. Figure 1C:

The pore profiling shows that they determined a closed state of the calcium channel. Instead of presenting 1.0-Å pore radius, I believe that presenting a more relevant pore radius such as a calcium ion radius (dehydrated and hydrated) can improve this panel.

5. Supplementary Figure 3.

This figure needs improved labeling to give more information to readers. This figure presents evidence to support the authors' statements in lines 142 - 174, thus should corresponding labelings should be presented here. For example, the addition of 'S4-S5' labeling will help readers in understanding what they are saying in lines 150-152.

6. EM validation report

Magnification information in the EM validation report is different from Supplementary Table 1 and the method part. Please correct it.

1 Point-by-point response for

2 Molecular insights into the gating mechanisms of voltage-gated calcium channel Cav2.3

TO REVIEWER REPORT #1

*The manuscript by Y Gao et al “Molecular insights into the gating kinetics of voltage-gated calcium channel Cav2.3”*
*is devoted to structural and functional study of Cav2.3 channel. The topic of study is actual; the work has excellent*
*novelty and executed using state-of-the-art methodology. To my opinion, the manuscript could be accepted for*
*publication in Nature Communications after minor revision.*

*There are several major and lot of minor concerns, which should be addressed by the authors before the acceptance.*

0 **Reply:** We greatly appreciate the reviewer’s positive comment.

Major points

*1) When describing obtained functional/mutagenesis results, the authors often use the term “kinetics of*
*gating/inactivation”. However, in most cases, they did not study the kinetics (the rate of processes), they simply measured*
*some properties of the activation/inactivation processes, e.g. shift of the activation curve along the voltage axis. The*
*obtained data give information about energy changes during transitions, but do not provide the information about the*
*kinetics of gating process.*

**Reply:** We appreciate this comment. We have re-evaluated all ‘kinetics’ mentioned in the manuscript and have
substituted some inappropriately used ‘kinetics’ with other words such as property, profile or mechanism.

0 Now they read:

(Title) Molecular insights into the gating mechanisms of voltage-gated calcium channel Cav2.3.

(Line 86) However, the inactivation properties of Cav1 and Cav2 channels are dramatically different³³.

(Line 202) Moreover, we tested whether the VSD_{II}^{4Q} mutant has a distinct CSI profile.

(Line 248) These alterations on the CSI profile suggested that the positively charged R781, R786 and K787 are critical
for the CSI mechanism of Cav2.3.

(Line 301) In particular, they displayed a ~7-mV negative shift on inactivation curve (Figure 4h), enhanced cumulative
inactivation to AP trains (Figure 4i and 4k), and an accelerated recovery rate from CSI (Figure 4j and Supplementary
Figure 6b), highly identical to the CSI profile of the $\Delta pre-w-helix$.

*2) The manuscript contains very large number of typos and small mistakes, including reference to the figures, mutant*
*naming etc.*

**Reply:** We thank the reviewer(s) for point out this. The manuscript has now been thoroughly checked again, and we
have corrected all the typos and mistakes that we found and pointed out by the reviewers. Please also see the responses
below.

*3) Many figures are drawn using pale colors and too small for understanding. They should be either redrawn using*

*more contrast palette or substantially increased in size.*

**Reply:** We thank the reviewer for this comment. We have enlarged Figure 1, 2 and 3. The schematic diagrams of figure
change in size are attached here for your convenience.

*4) Some important experimental details about sample preparation (e.g. detergent composition) are missing.*

**Reply:** Details for sample preparation has been clarified in the revised manuscript. Please also see replies to minor
points 3, 4, 44, 45, and 46.

Now they read:

(line 101) The Cav2.3- α 2 δ 1- β 1 complex was solubilized using n-Dodecyl- β -D-maltoside (DDM) and purified using a
strep-tactin affinity column, followed by further purification by size-exclusion chromatography (SEC) in a running buffer
containing glycol-diosgenin (GDN) to remove protein aggregates (Supplementary Figure 1a, see Method section for
details).

(line 408) The membrane was resuspended again using Buffer W and solubilized by the addition of 1% (w/v) n-dodecyl-
β -D-maltoside (DDM) (Anatrace, USA), 0.15% (w/v) cholesteryl hemisuccinate (CHS) (Anatrace, USA), 2 mM adenosine
triphosphate (ATP) and 5 mM MgCl₂ on a rotating mixer at 4°C for 2 h. Addition of ATP and MgCl₂ is to remove associated
heat shock proteins.

(line 417) The purified Cav2.3 complex was eluted using 15 mL elution buffer (20 mM HEPES pH 7.5, 150 mM NaCl, 5
mM β -ME, 0.03% (w/v) GDN, and 5 mM d-Desthiobiotin (Sigma-Aldrich, USA)) and concentrated to 1 mL using a 100
33 kDa MWCO Amicon (Millipore, USA).

(line 420) The concentrated protein sample was subjected to further size-exclusion chromatography (SEC) by a
Superose 6 Increase 10/300 GL gel filtration column (GE Healthcare, USA) using a flow rate of 0.3 mL/min and a running
buffer containing 20 mM HEPES pH 7.5, 1.5 mM NaCl, 5 mM β -ME, and 0.01% (w/v) GDN (Anatrace, USA).

*5) The data for W778 mutant is required.*

**Reply:** We thank the reviewer for this comment and agree with reviewer that we should study the gating properties of
W778 mutation. Here, we substitute the W778 by a hydrophilic glutamine (Cav2.3^{W/Q}) to disrupt hydrophobic interactions.
It leads to similar changes in gating properties as the Δw -helix deletion, namely unaltered activation curve, positively

i1 shifted inactivation curve and suppressed CSI, in line with our previous findings on Cav2.2^[1]. We have incorporated the
i2 related data in the revised manuscript.

i3 Related discussions are provided in the revised line 234. It now reads: "First, we designed the Cav2.3^{W/Q} (W778Q) to
i4 disrupt interactions between the W-helix and intracellular gate (Figure 3c–3g and Supplementary Figure 6). It turns out
i5 that the Cav2.3^{W/Q} exhibited a ~8-mV positive shift on the steady-state inactivation curve (Figure 3d) and an alleviated
i6 cumulative inactivation in response to AP trains (Figure 3e and 3g) without affecting the voltage dependence of channel
i7 activation (Figure 3c), consistent with the observations in Cav2.2²⁸, suggesting that the W778 is important for CSI
i8 process of Cav2.3 channel."

i9 The revised Figure 3 is attached below for your convenience.

0

Reference:

[1] Dong, Yanli, et al. "Closed-state inactivation and pore-blocker modulation mechanisms of human Cav2.2." *Cell*
*Reports* 37.5 (2021): 109931.

**Detailed description and minor points:**

*1) Introduction section mentions that there are several types of Cav channels L-, N-, T-, etc. Almost all the channels are*

*attributed to some type, but not the Cav2.3 itself. Please provide the type of this channel (R-) and briefly describe the*
*major biophysical properties of this type of channels.*

**Reply:** We appreciate this comment and have provide biophysical profiles of Cav2.3 in the introduction section at the
line 52. It now reads: “The so-called pharmacoresistant (R-type) Cav2.3 is widely expressed in the brain and enriched
in the hippocampus, cerebral cortex, amygdala, and corpus striatum¹⁰⁻¹². Electrophysiological investigations revealed
that currents mediated by Cav2.3 are resistant to common Cav blockers or gating modifiers such as nifedipine,
nimodipine, ω -Aga-IVA, etc¹³. Cav2.3 channels exhibit cumulative inactivation in response to brief and repetitive
depolarizations, a process known as preferential closed-state inactivation (CSI)¹⁴.”.

*2) Introduction, line 77. Abbreviation CSI is used before its introduction at line 78.*

**Reply:** We thank the reviewer for this comment and have provided the spelled-out term of CSI the first time it appears
at the line 56. It now reads: “Cav2.3 channels exhibit cumulative inactivation in response to brief and repetitive
depolarizations, a process known as preferential closed-state inactivation (CSI)¹⁴.”.

*3) Results and Discussion, lines 95-105. Please provide the detergent composition of the sample.*

**Reply:** We have provided the detergent composition in the revised Results and Discussion section. Detailed information
of the buffers has also been clarified in the Method section at the line 101. It now reads: “The Cav2.3- α 2 δ 1- β 1 complex
was solubilized using n-Dodecyl- β -D-maltoside (DDM) and purified using a strep-tactin affinity column, followed by
further purification by size-exclusion chromatography (SEC) in a running buffer containing glycol-diosgenin (GDN) to
remove protein aggregates (Supplementary Figure 1a, see Method section for details).”.

*4) Figure S1. Please provide details of SEC (detergent and buffer composition, column, flow rate).*

**Reply:** We appreciate the reviewer’s comment. We have provided the experimental details in revised Method section
in the line 420, it reads “The concentrated protein sample was subjected to further size-exclusion chromatography (SEC)
by a Superose 6 Increase 10/300 GL gel filtration column (GE Healthcare, USA) using a flow rate of 0.3 mL/min and a
running buffer containing 20 mM HEPES pH 7.5, 1.5 mM NaCl, 5 mM β -ME, and 0.01% (w/v) GDN (Anatrace, USA).”.

*5) Figure S1 caption. β 1 instead of δ 1.*

**Reply:** We appreciate the reviewer’s comment. We have corrected this typo. Now it reads (Figure S1 caption): “...
Bands representing the Cav2.3 α 1E, α 2 δ 1, and β 1 subunits were labeled...”.

*6) Figure S2 caption. “Heterologous refinement and non-uniform (NU) refinement in generated the final map, which*
*was reported at 3.1 Å” something missing after “in”.*

**Reply:** We thank the reviewer for pointing this out. We have removed the “in”. In the revised Figure S2 caption, it now
reads “Heterologous refinement and non-uniform (NU) refinement generated the final map, which was reported at 3.1
Å according to the golden-standard *Fourier* shell correlation (GSFSC) criterion.”.

*7) Figure S2 caption. S6II instead of S4II.*

**Reply:** We thank the reviewer for pointing this out. We have revised Figure S2 caption, it now reads: “Representative
cryo-EM density map (transparent grey surface) of the Domain II. Residues on the S4_{II}, S6_{II}^{NCD} and W-helix were
labeled.”.

*8) Line 114, please mention the helices that surround ECLs. (e.g. ECLs located between S5, P1, P2, or S6 helices in*
*pore domain).*

**Reply:** We thank the reviewer for pointing this out and have incorporated these details in the revised manuscript. In the

revised line 117, it now reads “Similar to other Cav channels, Cav2.3 harbors four extracellular loops (ECLs) that is also
positioned between S5 and S6 helices in the pore domain (Figure 1a–1c).”.

*9) Line 120, please repeat once more that W-helix belongs to D2-D3 linker.*

**Reply:** Thanks for your kind reminder. We have made adjustment as you suggested. In the revised line 123, it now
reads “Moreover, the closed gate of Cav2.3 is further stabilized by the W-helix from the DII-DIII linker, which is consistent
with a previous study on Cav2.2 and indicates that Cav2.3 also adopts the CSI mechanism²⁸.”.

*10) Lines 131-134, the reference to figure 1e is missing.*

**Reply:** We thank the reviewer for pointing out this. Reference to Figure 1e has been added. In the revised line 138, it
now reads: “Structural comparison of Cav2.3 and ziconotide-bound Cav2.2 demonstrated that the ECL_I loop of Cav2.3
adopts a different conformation, and residues D263 and P264 are placed close to the central axis, giving rise to clashes
between the ziconotide and the ECL_I of Cav2.3 (Figure 1e).”

*11) Lines 139-141, incorrect figure reference. Should be 1f instead of 1d.*

**Reply:** We thank the reviewer for pointing out this typo and have corrected it in our revised manuscript.

*12) Line 141, the literature reference(s) supporting the statement about shift in the half-activation voltage is needed.*

**Reply:** Thank you very much for this comment. The literature reference supporting this statement^[1] have been provided
in the revised manuscript. In the line 149 it now reads: “Eight of thirteen mutations are located around the intracellular
gate, such as I603L, F698S, and I701V, and result in a hyperpolarizing shift in the half-activation voltage²⁴.”.

Reference:

[1] Helbig, Katherine L., et al. "De novo pathogenic variants in CACNA1E cause developmental and epileptic
encephalopathy with contractures, macrocephaly, and dyskinesias." *The American Journal of Human Genetics* 103.5
(2018): 666-678.

*13) Figure 1, panel f is unclear. It should be redrawn using bright colors, or whole figure 1 should be resized.*

**Reply:** The Figure 1 have been resized to 2 times its original size and Figure 1f were redrawn using bright colors
according to your suggestion. The revised figure is attached below for your convenience.

8

9 *14) Lines 167,170. Loop instead of loops.*

0 **Reply:** These typos have been corrected.

Now they read:

(line 178) "... to disrupt the contacts between ECL_{IV} and S1-S2_{III} loop."

(line 181) "We thus speculate that the interactions between ECL_{IV} and the S1-S2_{III} loop may stabilize the VSD_{III} in a
certain conformation relative to the pore domain that requires less electrical energy to activate the channel, ...".

*15) Line 171. Incorrect use of term "gating kinetics". Provided data do not give any information about kinetics.*

**Reply:** We agree with the reviewer. This sentence is rephrased at your suggestion (Minor #16).

*16) Line 170. Too complicated speculation about influence of ECL on coupling of the VSD to the gate. Probably situation
is simpler. ECL stabilize the VSD in some conformation, that change it probability to switch in the activated state
(requiring less electric force). Please describe this. The gate of the channel is too far from the ECL.*

0 **Reply:** We agree with the reviewer and have rephrased our statement according to this comment. In the line 181, it now
reads "We thus speculate that the interactions between ECL_{IV} and the S1-S2_{III} loop may stabilize the VSD_{III} in a
certain conformation relative to the pore domain that requires less electrical energy to activate the channel, reminiscent of the
cholesterol regulation...".

*17) Line 174. The cholesterol/lipid molecules shown on the figure 2a is not described in the main text or figure caption.*

**Reply:** We have added description for the cholesterol hemisuccinate in the caption of Figure 2. It now reads: "a. ...
Cholesteryl hemisuccinate (CHS) molecules are shown as sticks and labeled."

i7 18) Figure S3. Please show the W-helix and pre-W-helix (if it observable in your structure) on the panels b-d.

i8 **Reply:** The W-helix have been shown in the Figure S4b–S4d (previously S3b–S3d) in the revised submission. The
i9 revised Figure S4b–S4d are attached here.

i0

i1 19) Figure 2. The trace for the WT channel should be added to the panel d.

i2 **Reply:** Representative trace for Cav2.3 WT have been added to the panel d in the revised submission. The revised
i3 Figure 2d is attached below.

i4

i5 20) Caption to the figure 2. The “minus” signs are missed before 100 and 15 mV. (-100 and -15)

i6 **Reply:** We thank the reviewer for pointing out this typo. This typo has been corrected. The revised Figure 2 caption now
i7 reads “... cells were stepped from a holding potential of -100 mV to pre-pulse potentials between -100 and -15 mV in
i8 5-mV increments for 10 s.”.

i9 21) Line 185. Mutant instead of mutants.

0 **Reply:** We thank the reviewer for pointing out this typo and we have corrected it in our revised manuscript.

*22) Lines 192-193. The conclusion about the alternative conformation of VSD-4Q is obvious, but how this state differs*
*from the resting state is unknown. I propose to change this conclusion to the “conformation of VSD-II influences gating*
*of the channel”.*

**Reply:** We agree with the reviewer and have changed the conclusion in the revised manuscript at the line 204. It now
reads: “These results suggested that the conformation of VSD_{II} influences the gating of Cav2.3.”.

*23) Line 194. Kinetics of OSI on the figure S4a was not analyzed. Please either provide results of R200 analysis (like in*
*Figure S7) or rephrase.*

**Reply:** R200 analysis of the wild-type Cav2.3 and VSD_{II}^{4Q} have been added to the Supplementary Figure 5d and is
attached below.

*24) Lines 196-199. This statement represents major result of the corresponding section. Please consider variant to move*
*it in the beginning of the section. The fact that VSD-2 influences (not controls) the voltage-gated activation should be*
*mentioned. For the VSD-4Q mutant you see not only CSI changes, but voltage shift of the activation curve.*

**Reply:** We appreciate this comment and agree with reviewer that mutations on the VSD_{II} influence the activation and
inactivation of the channel. We have adjusted this sentence in the revised manuscript at the line 209, it now reads that
“... while the VSD_{II} is not necessary for channel activation by sensing the depolarization of membrane potential; instead,
the VSD_{II} is crucial to modulating channel properties, such as CSI and voltage dependency of channel activation and
inactivation.”.

*25) Figure S4. The caption describing panel c is unclear. Please provide more details for measuring AP trains in*
*hippocampus and mention that the responses of mutated Cav2.3 channels were recorded in the HEK cells overexpressing*
*the channels.*

**Reply:** The AP trains used to repetitively activate Cav2.3 were recorded using a whole-cell current-clamp from a mouse
hippocampal CA1 pyramidal neuron after current injection. The detailed method was reported in a previous publication
24 ^[1]. We have clarified this in the figure legend and the Method section and added the reference.

Now they read:

(line 501) “The AP trains used to stimulate the HEK 293-T cells were recorded from a mouse hippocampal CA1 pyramidal
neuron after current injection in the whole-cell current-clamp mode⁵⁷.”.

(Figure S5c caption): “Representative current responses stimulated by action potential (AP) trains (left). The AP trains
were recorded using a whole-cell current-clamp from a mouse hippocampal CA1 pyramidal neuron after current injection
(see Method section for the literature reference).”.

Reference:

[1] Liu, Yongqing, et al. "CDYL suppresses epileptogenesis in mice through repression of axonal Nav1.6 sodium channel
expression." *Nature Communications* 8.1 (2017): 1-17.

26) *Figure S4. Data for 4G mutant should be added to the panel d.*

**Reply:** Electrophysiological data of Cav2.3^{4G} were added to Figure S5d. The revised Figure S5d is attached below.

27) *Figure S4d. Please clarify which statistic test was used.*

**Reply:** We thank the reviewer for this comment. The statistic test was clarified in the revised Supplementary Figure 5
caption. It now reads: "... Significances were determined using two-sided, unpaired *t*-test ...".

28) *Lines 222-223 and Figure 3. The panel 3a is too small and not bright. Please, consider other colors or resize the
panel. The full gray surface of alpha subunit is unnecessary.*

**Reply:** The gray surface of alpha subunit is removed, and the size of Figure 3 is enlarged in the revised submission.
The revised Figure 3a is attached below.

29) *Lines 226, 231. K787 should be instead of R787.*

**Reply:** We thank the reviewer for pointing out our oversights. The typos have been corrected.

Now they read:

(line 242) "To evaluate our speculations, we constructed the Cav2.3^{RK/A} mutant by substituting the R781, R786 and K787
with alanine (R781A/R786A/K787A)."

(line 248) "These alterations on the CSI profile suggested that the positively charged R781, R786 and K787 are critical

for the CSI mechanism of Cav2.3.”.

*30) Line 227. Please mention, important observation, that activation of the channel is not changed by RK/A mutation.*

**Reply:** We agree with the reviewer and have emphasized that the activation property is unaltered in the revised
manuscript at the line 244. It now reads: “The activation curve of the Cav2.3^{RK/A} mutant remains unaltered (Figure 3c).”.

*31) Lines 219-231. It is very interesting what happens upon mutation of W778 residue. This data is needed to support*
*conclusions of the work.*

**Reply:** We appreciate this comment and agree with reviewer that we should study gating properties of W778 mutation.
Here, we substitute the tryptophan by hydrophilic glutamine (Cav2.3^{W/Q}) to disrupt hydrophobic interactions. It leads to
similar changes in gating properties as Δw -helix deletion, namely unaltered activation curve, positively shifted
inactivation curve and suppressed CSI, in line with our previous findings on Cav2.2^[1]. We have incorporated these data
in the revised manuscript.

Related discussions are provided in the revised line 234. It now reads: “First, we designed the Cav2.3^{W/Q} (W778Q) to
disrupt interactions between the W-helix and intracellular gate (Figure 3c–3g and Supplementary Figure 6). It turns out
that the Cav2.3^{W/Q} exhibited a ~8-mV positive shift on the steady-state inactivation curve (Figure 3d) and an alleviated
cumulative inactivation in response to AP trains (Figure 3e and 3g) without affecting the voltage dependence of channel
activation (Figure 3c), consistent with the observations in Cav2.2²⁸, suggesting that the W778 is important for CSI
process of Cav2.3 channel.”.

Reference:

[1] Dong, Yanli, et al. "Closed-state inactivation and pore-blocker modulation mechanisms of human Cav2.2." *Cell*
*Reports* 37.5 (2021): 109931.

The data is shown in the revised Figure 3 and we have included the figure here for your convenience.

*32) Line 248. "kinetics of channel activation" was not measured.*

**Reply:** We thank the reviewer for pointing out this and have removed the 'affecting the kinetics' in the revised sentence.
 In the line 265, it now reads: "The Δw -helix exhibited a ~ 4 -mV positive shift on the steady-state inactivation curve (Figure
 4c) and alleviated cumulative inactivation in response to AP trains (Figure 4d and 4k) without affecting the voltage
 dependence of channel activation (Figure 4b), indicating that the W-helix plays pivotal roles in the CSI of $Ca_v2.3$ ".

*33) Figures 3 and 4 contain similar data and identical curves for WT and -dW-helix channels. Please consider to join
 these figures in one. The source data for RK/A mutant should be shown, like in the panel 4k.*

**Reply:** We thank the reviewer for this comment. Figures 3 and 4 differently focus on CSI modulation by the W-helix and
 the pre-W-helix, respectively. So, we divide these data into two figures to clearly display data and facilitate reading.

Representative traces for $Ca_v2.3^{RK/A}$ have been provided in Figure 3g in the revised manuscript. The adapted Figure 3g
 are attached below. Please also see complete Figure 3 in reply to comment #31.

34) *Figure 4. Caption to panel k is unclear; please see above. The reference to Figure S5 should be changed to the reference to Figures S4 and S5.*

Reply: We thank the reviewer for pointing out this. Caption to panel k have been revised. It now reads: “Representative current responses to the AP trains. See Supplementary Figure 5 for voltage-clamp protocols and Method section for literature reference.”.

Figure references have been updated in the revised manuscript. In the line 299 it now reads: “... but significantly different from that of the WT $Ca_v2.3$ channel (Figure 4g–4k, Supplementary Figure 5a–5b and Supplementary Figure 6b).”.

We have also clarified the experimental details of AP trains in the revised Method section. In the line 501, it now reads: “The AP trains used to stimulate the HEK 293-T cells were recorded from a mouse hippocampal CA1 pyramidal neuron after current injection in the whole-cell current-clamp mode⁵⁷.”.

35) *Figure 4. It is very inconvenient to show only one legend (with mutant/coloring encoding) for such a large multi-panel figure. Please consider variant to provide separate legends for panels b-e and g-j.*

Reply: We thank the reviewer for this comment. We have added more legends for color coding as reviewer suggested. The revised figure is attached below for your convenience.

0

36) Figure S5. The panel a is redundant, it repeats the figure S4c (left). Please remove.

Reply: The original panel a has been removed in the revised Supplementary Figure 6.

37) Line 279. Not only kinetics but other properties are similar for preW/Q and preR/Q mutants. Please mention it.

Reply: We agree with the reviewer and have adjusted the sentence according to your suggestion. In the line 297, it
 reads: "Impressively, the mutants Ca_v2.3^{preR/Q} and Ca_v2.3^{preW/Q} exhibit similar gating kinetics and voltage dependence
 of channel activation and inactivation, but significantly different from that of the WT Ca_v2.3 channel (Figure 4g–4k,
 Supplementary Figure 5a–5b and Supplementary Figure 6b).".

38) Line 286. Figure 4d and 4i instead of I and k.

Reply: We thank the reviewer for pointing out this. The figure reference has been updated.

39) Line 287. Figure 4e instead of Figure 4.

Reply: We thank the reviewer for pointing out this. This figure reference has been updated.

40) Line 335. Figure 5c instead of 5d.

Reply: We thank the reviewer for pointing out this. This figure reference has been updated.

41) Line 345. The sentence "S6IINCD plays central roles in connecting with both AID and the S4-S5 helix." is

*unsupported conclusion and not connected with surrounding text fragments.*

**Reply:** We agree with the reviewer and have removed this sentence in the revised manuscript.

*42) Line 364. Please mention that R590 residue belongs to S4-S5 linker.*

**Reply:** We appreciate this comment and have mentioned it in the revised manuscript. In the line 381, it now reads:
“Conclusions drawn using time constants are nearly identical to those using R200 values (Supplementary Figure 8d),
further supporting that S6_{II}^{NCD}, R371 (AID), R378 (AID), and R590 (S4-S5_{II}) play important roles in OSI modulation.”

*43) Figure 5d. Please clarify, which statistic test was used.*

**Reply:** We thank the reviewer for point out this. Two-sided, unpaired *t*-test were used to determine the significances. We
have clarified this in the revised Figure 5d caption, it now reads: “... Significances were determined using two-sided,
unpaired *t*-test ...”.

*44) Lines 389-390. Please provide description, why ATP and MgCl₂ are needed for channel purification.*

**Reply:** We appreciate this comment. When the Cav channel complex is recombinantly expressed in HEK293 cell, heat
shock protein 70 (HSP70) is likely essential for protein correctly folding. Its expression is also enhanced during Cav
channel overexpression. However, a part of HSP70 is usually co-eluted with the Cav channel. To remove HSP70
contamination in the purified Cav sample, we introduced the ATP and MgCl₂ in our purification, because Mg-ATP induce
conformational transition of HSP70 and results in the dissociation of HSP70 from the substrate polypeptides^[1].

We have clarified this in the revised Method section. In the line 408, it now reads: “The membrane was resuspended
again using Buffer W and solubilized by the addition of 1% (w/v) n-dodecyl-β-D-maltoside (DDM) (Anatrace, USA), 0.15%
(w/v) cholesteryl hemisuccinate (CHS) (Anatrace, USA), 2 mM adenosine triphosphate (ATP) and 5 mM MgCl₂ on a
rotating mixer at 4°C for 2 h. Addition of ATP and MgCl₂ is to remove associated heat shock proteins.”.

Reference:

[1] Palleros, Daniel R., William J. Welch, and Anthony L. Fink. "Interaction of hsp70 with unfolded proteins: effects of
temperature and nucleotides on the kinetics of binding." *Proceedings of the National Academy of Sciences* 88.13 (1991):
5719-5723.

*45) Lines 396-397. Please provide composition of elution buffer. (including detergent composition).*

**Reply:** We appreciate this comment and have provided the composition of the elution buffer in the revised Methods
section at the line 417. It now reads: “The purified Cav2.3 complex was eluted using 15 mL elution buffer (20 mM HEPES
pH 7.5, 150 mM NaCl, 5 mM β-ME, 0.03% (w/v) GDN, and 5 mM d-Desthiobiotin (Sigma-Aldrich, USA)) and
concentrated to 1 mL using a 100 kDa MWCO Amicon (Millipore, USA).”.

*46) Lines 398-400. Please provide composition of buffers used for SEC. (including loading buffer and elution buffer).*

**Reply:** We thank the reviewer for this comment. Composition of the SEC running buffer have been provided in the
revised Method section at the line 420. It now reads: “The concentrated protein sample was subjected to further size-
exclusion chromatography (SEC) by a Superose 6 Increase 10/300 GL gel filtration column (GE Healthcare, USA) using
a flow rate of 0.3 mL/min and a running buffer containing 20 mM HEPES pH 7.5, 1.5 mM NaCl, 5 mM β-ME, and 0.01%
(w/v) GDN (Anatrace, USA).”.

!1 **TO REVIEWER REPORT #2**

!2 *Yiwei Gao and collaborators present the first high-resolution (3.1 Angstrom) cryo-EM structure of the human R-type*
!3 *voltage gated calcium channel CaV2.3. The structure of the pore forming subunit alpha1E is resolved in complex with*
!4 *the accessory subunits alpha2delta1 and beta1 providing the original information about the organization of this*
!5 *macromolecule. CaV2.3 pore is in the closed state stabilized by interactions with the PreW- and W- helices, suggesting*
!6 *a mechanisms for closed-state inactivation. The voltage sensors are in the active position with the exception of VSDII*
!7 *in which the S4 segment is held into its resting position by specific interactions with an unknown molecule.*

!8 *By comparing CaV2.3 and Cav1.1 pore regions, the Authors identified two residues that pose steric hindrance to*
!9 *nifedipine binding, thus providing a reasonable structural basis for the DHP resistance of Cav2.3. Similarly, the*
!10 *structural basis for Cav2.3 resistance to ziconotide is revealed by comparison with a ziconotide-bound Cav2.2 structure.*

!11 *Overall, the presented structural work is valuable and has the potential to advance the field: it contributes a high-*
!12 *resolution structure of a new voltage gated calcium channel, shedding light on key mechanisms of channels gating and*
!13 *pharmacoresistance.*

!14 *However, I have several concerns about functional studies included in this manuscript: they should be addressed. I also*
!15 *find that some of the conclusions are not supported by specific data (e.g. lack of experimental results supporting the*
!16 *proposed role of the voltage sensing domains).*

!17 *Some of the statements require qualification and/or references.*

!18 *There are many mistakes and typos.*

!19 **Reply:** We appreciate very much the reviewer's positive comment and his/her suggestions for improving our manuscript.

!1 **Main points**

!2 *1. The authors want to assign a role to the interaction between the ECL-IV and the VSDIII. Are G/Gmax curves for WT*
!3 *and 4G (Fig2 B) significantly different?*

!4 *What are the evidences that CaV3.2 VSDIII is energetically coupled to the pore?*

!5 *The 5 mV shift in the GV curve as consequence of the 4G mutation in ECL-IV may be due to a destabilization of the*
!6 *open state and not a change in coupling of VSDIII, as suggested. The slope of Cav2.34G G/Gmax curve should decrease*
!7 *when a VSD that contributes to voltage dependent activation is partially uncoupled from the pore, as it is proposed for*
!8 *the 4G mutant. Instead, it looks like Cav2.3^{4G} produce a parallel shift of the G/Gmax curve (same slope as WT G/Gmax)*
!9 *(Fig. 2B).*

!10 *In addition, to fully establish a functional role for these interactions, mutations in S1-S2-III liker (V, L, T, N residues)*
!11 *should be tested: in principle, they should cause similar changes in channel voltage dependence.*

!12 **Reply:** We appreciate this comment. The ~5-mV positive shift in activation curve of Cav2.3^{4G} is statistically significant
!13 (P < 0.0001, two-tailed unpaired t-test). We have incorporated this in the revised manuscript. In the line 179, it reads
!14 "Electrophysiological studies indicated that the voltage dependency of the activation curve of Cav2.3^{4G} displayed a ~5-
!15 mV positive shift (P < 0.0001, two-tailed unpaired t-test) compared to that of wild-type Cav2.3 (Figure 2b and
!16 Supplementary Figure 5).".

!17 In our study, the VSDIII^{5Q} mutant failed to elicit Ca²⁺ current, suggesting the VSDIII may be important for channel function.
!18 Moreover, previous studies have also demonstrated that VSDIII of Cav2.3 or the closely-related Cav2.2 play vital roles

in gate opening [1-2].

As reviewer mentioned, the slopes of G/G_{max} curve of $Ca_v2.3$ WT and $Ca_v2.3^{4G}$ are nearly identical. However, previous studies suggested that the slope of the activation curve is proportional to the amount of gating charge [3-4].

We agree with reviewer in that it would be ideal if we can reach a similar conclusion from mutations in S1-S2_{III} linker. Therefore, we constructed another mutant by mutating the ¹¹⁷⁶VLN¹¹⁷⁹ to four glycines (VSD_{III}^{4G}) as reviewer suggested and carried out electrophysiological experiments (panel a-b). However, the current density mediated by VSD_{III}^{4G} is severely reduced, relative to both WT $Ca_v2.3$ and $Ca_v2.3^{4G}$ mutant (panel c), suggesting that the S1-S2 loop is important for channel function. Most importantly, the functional properties of $Ca_v2.3^{4G}$ are apparently distinct from those of VSD_{III}^{4G} , suggesting that the roles of the ECL_{III} loop and S1-S2 loop in channel function are not completely identical. We speculate that the S1-S2 loop may be involved in the modulation of channel function in addition to its interaction with ECL_{III}. However, the possibility that the ECL_{III} loop interacts with the S1-S2 loop to participate in channel function cannot be ruled out.

According to reviewer-1's comment #16, we revised discussions about $Ca_v2.3^{4G}$. In the line 181, it now reads "We thus speculate that the interactions between ECL_{IV} and the S1-S2_{III} loop may stabilize the VSD_{III} in a certain conformation relative to the pore domain that requires less electrical energy to activate the channel, reminiscent of the cholesterol regulation...".

a. Voltage-clamp protocol for activate curve determination. b. Representative current traces for the $Ca_v2.3$ WT and VSD_{III}^{4G} . c. Current density of the $Ca_v2.3$ WT, $Ca_v2.3^{4G}$, and VSD_{III}^{4G} .

References:

- [1] Bourinet, Emmanuel, et al. "Interaction of SNX482 with domains III and IV inhibits activation gating of $\alpha 1E$ ($Ca_v2.3$) calcium channels." *Biophysical Journal* 81.1 (2001): 79-88.
- [2] Chai, Zuying, et al. " $Ca_v2.2$ gates calcium-independent but voltage-dependent secretion in mammalian sensory neurons." *Neuron* 96.6 (2017): 1317-1326.
- [3] Yifrach, Ofer, and Roderick MacKinnon. "Energetics of pore opening in a voltage-gated K^+ channel." *Cell* 111.2 (2002): 231-239.
- [4] Zizi, Martin, et al. "NADH regulates the gating of VDAC, the mitochondrial outer membrane channel." *Journal of Biological Chemistry* 269.3 (1994): 1614-1616.

2. Please, in Fig.2C, display the IV curve for Cav2.3-4G together with WT and VSDII-4Q.

**Reply:** The I-V curve for Cav2.3^{4G} has been displayed together with the wild-type Cav2.3 and VSDII^{4Q}.

3. Conclusions regarding the role of VSDII and the 4Q mutant are vague and unclear.

*Line 192: the Authors conclude that “the VSDII of VSDII4Q mutant may adopt an alternative conformation other than the resting state in the wild-type Cav2.3.” This seems an important point of the study; however, this “alternative conformation” is not explained. Likewise, how this alternative conformation relates to the observed effect on inactivation and voltage dependence is not discussed.*

**Reply:** We appreciate this comment. In previous studies, researchers had investigated the functional roles of VSDs by neutralizing gating charge on the S4 helices^[1-2]. However, the conformation of the neutralized VSDs is still under debate. Both activated ‘up’ conformation and resting ‘down’ conformation have been proposed in previous studies^[2-3].

In our study, the VSDII^{4Q} mutant exhibits ~9-mV left shifts in the voltage dependency of both activation and steady-state inactivation compared to the wild-type, suggesting the VSDII^{4Q} mutant requires less energy to activate the channel. The VSDII^{4Q} may be immobilized at an activated state. We further speculate that the S4_{II} helix at ‘up’ conformation leads to conformational changes in the S4-S5_{II} or even the S6_{II} helix, which stabilizes the binding of the W-helix to the intracellular gate and causes enhanced CSI. These are just our speculations and need further research to verify. We actually could not unambiguously define the conformation of the VSDII. To avoid overinterpreting, we prefer to leave this as an open question. We have adjusted the related discussion as reviewer-1 suggested (minor #22). In the line 204, it now reads: “These results suggested that the conformation of VSDII influences the gating of Cav2.3.”.

References:

[1] Cestèle, Sandrine, et al. "Neutralization of gating charges in domain II of the sodium channel α subunit enhances voltage-sensor trapping by a β -scorpion toxin." *The Journal of General Physiology* 118.3 (2001): 291-302.

[2] Capes, Deborah L., et al. "Domain IV voltage-sensor movement is both sufficient and rate limiting for fast inactivation in sodium channels." *Journal of General Physiology* 142.2 (2013): 101-112.

[3] Savalli, Nicoletta, et al. "The distinct role of the four voltage sensors of the skeletal Cav1.1 channel in voltage-dependent activation." *Journal of General Physiology* 153.11 (2021): e202112915.

3 4. Line 189: The Authors states that current density of the VSDII4Q mutant is “slightly increased” (Fig. 2C). Is this a statistically significant change? Why is it important to make this point? If the Authors think that this is due to the negative shift of $V_{1/2}$ of activation, perhaps they should say so, unless the mutant’s surface expression is higher than WT.

**Reply:** We thank the reviewer for this comment. The slight increase is a statistically significant change ($P < 0.0001$). We
agree with reviewer's interpretation that the difference in current density is caused by the negative shift of the activation
curve. We have clarified this in the revised manuscript at the line 198. It now reads: "Interestingly, the neutralization
mutation of the VSD_{II} (VSD_{II}^{4Q}) exhibits ~9-mV left shifts in the voltage dependency of both activation and steady-state
inactivation compared to the wild-type (Figure 2b and Supplementary Figure 5a–5b). Consequently, the current density-
voltage curve of the VSD_{II}^{4Q} mutant was also left shifted (Figure 2c).".

*5. Lines 347-348 regarding Open State Inactivation: the authors state that "CaV2.3N_Q mutant mediates a current that
decays much more slowly than the wild-type CaV2 suggesting that OSI is largely abolished". The Inactivation process
is still very prominent in the N_Q mutant thus claiming that OSI is "largely abolished" is not correct.*

**Reply:** We thank the reviewer for pointing out this. We have replaced "largely abolished" with "remarkably decreased"
at the line 365. It now reads: "The Cav2.3^{N_Q} mutant mediates a current that decays much more slowly than the wild-type
Cav2.3 during a 200-ms test pulse (Supplementary Figure 8a–8b), suggesting that OSI is remarkably decreased.".

*6. The neutralization of the charged residues in VSD I, III and IV did not elicit functional channels so no data are
presented about the properties of channel with the impaired VSDs (except for VSDII). The Authors state that "These
results indicate that the VSDI, VSDIII, and VSDIV play indispensable roles in voltage-gated pore opening". I do not
think such a conclusion can be made from non-functional channels (Fig. 2D).*

**Reply:** We thank the reviewer for this comment. We agree that the non-functional VSD_I^{5Q}, VSD_{III}^{5Q}, and VSD_{IV}^{5Q} are not
sufficient to fully support the conclusion. However, previous reports suggested that the gating charges of VSD_I were
regarded as the rate-limiting factor in the activation of close related Cav2.2^[1] and the VSD_{III}, and VSD_{IV} play vital roles
in the gating of Cav2.2^[1-3].

We have added these references and adjusted our statements in the revised manuscript.

In the line 206, it now reads: "In contrast, the gating charge-neutralized mutation in the VSD_I, VSD_{III}, and VSD_{IV} resulted
in failure to mediate inward current (Figure 2d), in line with previous results showing that the VSD_I, VSD_{III}, and VSD_{IV}
are important for gating of the closely-related Cav2.2³⁹⁻⁴¹...".

References:

[1] Zhong, Huijun, et al. "Control of gating mode by a single amino acid residue in transmembrane segment IS3 of the
N-type Ca²⁺ channel." *Proceedings of the National Academy of Sciences* 98.8 (2001): 4705-4709.

[2] Lin, Zhixin, et al. "Identification of functionally distinct isoforms of the N-type Ca²⁺ channel in rat sympathetic ganglia
and brain." *Neuron* 18.1 (1997): 153-166.

[3] Lin, Yingxin, Stefan I. McDonough, and Diane Lipscombe. "Alternative splicing in the voltage-sensing region of N-
Type Cav2.2 channels modulates channel kinetics." *Journal of Neurophysiology* 92.5 (2004): 2820-2830.

*7. Line 74: The statement that Cav2.3 channels inactivate from an intermediate closed state needs a reference(s).*

**Reply:** We thank the reviewer for this comment. Inactivation from an intermediate closed state is the hallmark of
preferential CSI^[1-2]. The references below have been added in the revised manuscript.

References:

[1] Patil, Parag G., David L. Brody, and David T. Yue. "Preferential closed-state inactivation of neuronal calcium
channels." *Neuron* 20.5 (1998): 1027-1038.

[2] McDavid, Sarah, and Kevin PM Currie. "G-proteins modulate cumulative inactivation of N-type (Cav2.2) calcium

channels." *Journal of Neuroscience* 26.51 (2006): 13373-13383.

8. Line 115: "In addition, the alpha2delta1 subunit interacts with the S1-S2I loop, implying that alpha2delta1 modulates
the gating kinetics of Cav2.3". Why does this interaction imply a modulation of gating kinetics? Please explain and
provides specific reference(s).

**Reply:** We thank the reviewer for point out this. Co-expressing $\alpha 2\delta 1$ with Cav2.3- $\beta 1$ shift both the activation and steady-
state inactivation curve towards a more polarized potential (-4 mV for activation curve, and -7 mV for inactivation curve)
compared to Cav2.3- $\beta 1$ alone [1]. Moreover, the negative shifts by $\alpha 2\delta 1$ have also been observed in other Cav1 and
Cav2 channels, such as Cav1.2 [1-2] and Cav2.2 [1,3]. However, we fully agree with the reviewer that the modulation effect
of $\alpha 2\delta 1$ on the channel may not be simply attributed to the interdomain interaction at the S1-S2I loop. In the revised
manuscript, this sentence is removed to avoid overinterpretation.

References:

[1] Yasuda, Takahiro, et al. "Auxiliary subunit regulation of high-voltage activated calcium channels expressed in
mammalian cells." *European Journal of Neuroscience* 20.1 (2004): 1-13.

[2] Felix, Ricardo, et al. "Dissection of functional domains of the voltage-dependent Ca²⁺ channel $\alpha 2\delta$ subunit." *Journal*
*of Neuroscience* 17.18 (1997): 6884-6891.

[3] Canti, C., et al. "The metal-ion-dependent adhesion site in the Von Willebrand factor-A domain of $\alpha 2\delta$ subunits is
key to trafficking voltage-gated Ca²⁺ channels." *Proceedings of the National Academy of Sciences* 102.32 (2005):
11230-11235.

9. The manuscript includes a full section entitled 'Functional heterogeneity of the voltage-sensing domains'. This is the
title of a published paper (Pantazis et al. PNAS 2014) that first demonstrated "functional heterogeneity" of Cav channel
voltage sensors: there is no reference to that (or related) studies.

**Reply:** We thank the reviewer for pointing out this. We have incorporated a new sentence in the revised submission
and have added this reference. In the line 163, it now reads: "Although the four VSDs of Cav channels are considerably
similar in terms of sequence and overall structure, they contribute differentially to the opening of pore³⁶".

10. Fig 4F is supposed to offer a model for the regulatory role of pre-W helix. As it is, the cartoon is not very helpful
and does not explain how Pre-W and W helices work to tune CSI. The inhibitory (or "competitive negative regulator")
role of Pre-w does not seem captured by the drawing.

**Reply:** We thank the reviewer for this comment. The Figure 4f was redrawn in the revised submission to emphasize the
'competitive negative regulator' role of Pre-W-helix.

*11. Line 327: The Authors mention the identification of a negatively charge domain, 715 EEQEEEE 721 in the*
*intracellular region of the S6II helix (Figure 5a–5b).*

*In the Cav2.3 structure (Fig. 5), residue 715 is Asp; If this is correct, the domain is DEQEEEE not EEQEEEE. This is*
*also in Figure 5 legend.*

**Reply:** We thank the reviewer for pointing this out. The typos have been corrected in the revised submission.

Now they read:

(line 346) “Interestingly, we identified a negatively charged domain ⁷¹⁵DEQEEEE⁷²¹ in the intracellular juxtamembrane
region of the S6_{II} helix (S6_{II}^{NCD}) (Figure 5a–5b).”.

(line 360) “... including replacing ⁷¹⁵DEQEEEE⁷²¹ with ⁷¹⁵NNQNNNN⁷²¹ (Cav2.3^{NQ}) ...”.

(Figure 5 caption) “... The S6_{II}^{NCD} (⁷¹⁵DEQEEEE⁷²¹) is shown as sticks.”.

*12. OSI-related mutants and Suppl. Fig 7. The representative current recordings for both R378Q and R378E clearly*
*display faster inactivation then WT. Nevertheless, in the plot of OSI ratios (R200, Suppl. Figure 7) R378Q appears to*
*be slower than WT.*

*Also, R200 values for R371E and R378E are practically identical in spite of the exceptional difference in their*
*inactivation kinetics shown on the recordings.*

*There is an obvious discrepancy between some of the representative current families and the estimated extent of*
*inactivation shown in the R200 plot. The simple R200 method does not seem adequate to describe the phenomena the*
*Authors are attempting to study here. Perhaps, a more accurate estimate of the inactivation properties using single or,*
*if necessary, double exponential fitting of the current decay, could provide more consistent results and facilitate their*
*interpretation.*

**Reply:** We thank the reviewer for this comment. However, in the Supplementary Figure 8b (previously 7b), R378Q
(brown trace) actually displayed R200 values lower (inactivated faster) than WT, consistent to the representative current
where R378Q decays faster. We have attached the representative traces and R200 values of WT and R378Q mutant
here for your convenience (**a**).

The time course of current decay could be well fitted by a single exponential (**b**). It turns out that the conclusions resulting
from the inactivation time constant agree well with those drawn from R200 value. Basically, the mutants exhibiting a
smaller R200 values shows a lower time constant.

We have added the exponential fitting analysis to the Supplementary Figure 8 and included a related discussion in the
revised manuscript at the line 381. It now reads: “Moreover, the time course of channel inactivation could be well fitted
by a single exponential. Conclusions drawn using time constants are nearly identical to those using R200 values
(Supplementary Figure 8d), further supporting that S6_{II}^{NCD}, R371 (AID), R378 (AID), and R590 (S4-S5_{II}) play important
roles in OSI modulation.”.

**a.** Representative traces of R378Q and CaV2.3 WT and their R200 analysis (extracted from
Supplementary Figure 8b and 8c). **b.** Single-term exponential fitting of the current traces of all
OSI-related mutants.

*Other points*

*1. Suppl. Fig.5B: in Cav2.3preR/Q, the peak current amplitude after 2s recovery (rightmost peak) is smaller than the*
*vertical arrow that represents (I am guessing) the peak inward current amplitude after a recovery time of 2048ms. Please*
*explain or correct.*

**Reply:** We thank the reviewer for pointing this out. This is a mistake that happened during figure preparation and have
been corrected in the revised Supplementary Figure 6b. The revised panel is attached here for your convenience.

*2. Suppl. Fig 1 b: Please indicate in the legend the expected MW of alpha1, alpha2delta and beta1*

**Reply:** We thank the reviewer for this comment. More details including the molecular weights for each band and other
labels have been incorporated in the revised Supplementary Figure 1b.

3. Line 284: *“the development and release of CSI is asynchronous”*. Should be *“are”*.

Reply: We thank the reviewer for this comment. This typo has been corrected in the revised version. Please see the reply to the next comment.

4. Line 284: *“Interestingly, our data also suggest that the development and release of CSI is asynchronous”*. It is not clear what asynchronous means here. Can development and release of CSI be *“synchronous”*? Please explain why this is an interesting finding.

Reply: We thank the reviewer for this comment. We have rephrased this sentence in the revised submission. In the line 305, it now reads: “Our data also show that the development and release of CSI are two independent processes.”.

5. Line 316: *“Although recent investigations have revealed the high-resolution cryo-EM structures of the Cav1.1 (L-type) and Cav2.2 (N-type) complexes, their imparity on OSI kinetics has not yet been explained”*. Not a clear sentence; please rephrase.

Reply: We thank reviewer for this comment and agree that this sentence should be rephrased to clarify our description. In the line 338, it now reads: “Although the structures of L-type Cav1.1²⁵ and N-type Cav2.2^{27,28} complexes have been elucidated at high resolution, the structural basis for their distinct OSI properties remains elusive.”.

6. Line 324: *“Previous studies suggested...”* but only a single study is reported (30).

Reply: We thank the reviewer for pointing out this and have added one more reference^[1] to this sentence. In the line 343, it now reads: “Previous studies suggested that the AID helix is able to regulate the OSI in Cav channels^{33,34}.”

Reference:

[1] Berrou, L., G. Bernatchez, and L. Parent. "Molecular determinants of inactivation within the I-II linker of $\alpha 1E$ (CaV2.3) calcium channels." *Biophysical Journal* 80.1 (2001): 215-228.

7. Line 334: *“which is consistent with structural observations that the AID of Cav1.1 has high motility (Figure 5d and Supplementary Figure 6).”* Figure 5d shows R200 ratios and does not seem relevant here.

Reply: We thank the reviewer for pointing out this. In the revised manuscript, this typo is substituted by ‘Figure 5c’, which shows the electrostatic profile of AID and S6_{II}. In the line 353, it now reads: “... which is consistent with structural observations that the AID of Cav1.1 has high motility (Figure 5c and Supplementary Figure 7).”.

8. Suppl. Fig 1 legend: *“delta 1”* should be *“beta 1”*.

Reply: We thank the reviewer for pointing out this typo. The revised Figure S1 caption now reads: “... Bands representing the Cav2.3 $\alpha 1E$, $\alpha 2\delta 1$, and $\beta 1$ subunits were labeled. ...”

9. Fig 4e: X axis is mislabeled: it should be "Time" not "Voltage".

Reply: We thank the reviewer for pointing this out and corrected this typo in the revised Figure 4e, which is attached below.

10. Line 195: "In contrast, the gating charge-neutralized mutation in the VSDI, VSDII, and VSDIV resulted in failure to mediate inward current". Should be "VSDI, VSDIII, and VSDIV."

Reply: We thank the reviewer for the comment and have corrected this typo.

11. Figure 5 title. "Structural basis of the modulation on OSI mechanism". Should be "of OSI mechanism".

Reply: We thank the reviewer for the comment and have corrected this typo.

TO REVIEWER REPORT #3

Voltage-gated calcium channels mediate calcium ion flux across the membrane of excitable cells. Among the calcium channels, Cav2.3 plays an important role in neurotransmitter release and is associated with neuronal dysfunctions such as epilepsy. However, there was little available structural information of Cav2.3 channel and thus understanding of the molecular mechanism of this channel has been hindered. In this paper, Gao et al. provided structural insights on pharmacology, gating kinetics, and inactivation of Cav2.3. They have determined a high-resolution cryo-EM structure of human Cav2.3 in complex with its auxiliary units.

The results provide the detailed mechanisms underlying the molecular function of Cav2.3 channel. The studies are straightforward, well-designed and the results are convincing and clearly described. The conclusions are supported by structural and functional approaches. But a few things need to be addressed before publication.

Reply: We greatly appreciate the reviewer's positive comments.

1. Line 99 - 101:

The authors already purified the complex and successfully determined the structure, therefore the statement for monodispersity and particle distribution in this part is unnecessary and misleading.

We cannot say that particles are distributed homogeneously with this result. The SEC trace displays that multiple species are in the peak and the peak seems a sum of several peaks with some high backgrounds. The author's statement here is refuted by the evidence they provided, Supplementary figure 1. In the supplementary figure 1b the SDS-PAGE shows many bands, so it is unlikely that protein particles were distributed homogeneously. In protein purification, it is not surprising that some minor impurities, aggregates and cleaved products are in a peak. And the minor impurities as well as some heterogeneous particles can be removed during EM data analysis. I believe that taking out this part does not

harm the excellence of this study.

Reply: We thank the reviewer for pointing out this and fully agree with the reviewer. The sentences at line 99–101 have been removed in the revised manuscript.

2. Supplementary Figure 1b:

Because of the multitude of protein bands, it is hard to know which bands are matched for Cav2.3 $\alpha 1E$, $\alpha 2\delta 1$, and $\beta 1$. Additional marks such as arrows or lines are necessary to help readers to understand the SDS-PAGE gel.

Reply: We thank the reviewer for this comment. Additional labels were added in the revised Supplementary Figure 1b, which is attached below. Bands representing each Cav_v2.3 components were labeled according to their molecular weights.

3. Line 122-134:

The Cav2.3 structure and comparing it with other Cav channels show structural basis underlying the pharmacoresistant property of Cav2.3. The Cav2.3 structure is of high quality and clearly shows why nifedipine and ziconotide cannot bind to Cav2.3. To further support this finding, relevant mutagenesis and functional studies testing nifedipine and ziconotide insensitivity are necessary. For example, mutations on D263 and P264 to make Cav2.3 sensitive to ziconotide will be of interest and can be supportive of their structural findings.

Reply: We appreciate this comment and agree with reviewer that it would be useful if we can confirm our observations about insensitivity of Cav_v2.3 to nifedipine and ziconotide. Therefore, we designed two constructs, including deletion of D263 and P264 (Ca_v2.3^{Zico}) and a double mutant F1708M/Y1296T (Ca_v2.3^{Nife}) and carried out electrophysiological experiments. Both constructs turn out to be functional (panel **a** and **c**), but still could not be inhibited by ziconotide (panel **a–b**) or nifedipine (panel **c–d**). We performed additional structural analysis and literature exploration in an attempt to fully explain why nifedipine and ziconotide cannot bind Ca_v2.3.

a. Representative current traces of $\text{Ca}_v2.3$ WT and $\text{Ca}_v2.3^{\text{Zico}}$ under 300 nM ziconotide treatment. **b.** Statistical analysis of the inhibition rate by ziconotide. **c.** Representative current traces of $\text{Ca}_v2.3$ WT and $\text{Ca}_v2.3^{\text{Nife}}$ under 500 nM nifedipine treatment. **d.** Statistical analysis of the inhibition rate by nifedipine. ns, not significant.

The structure of $\text{Ca}_v2.3$ is superimposed on the ziconotide-bound $\text{Ca}_v2.2$ structure (PDB ID: 7VFU). We found that in addition to the obvious clashes between ziconotide and D263-P264 ^{$\text{Ca}_v2.3$} , many other residues located on P-loops (P2_I, P2_{II}, P2_{III}, and P2_{IV}), ECL_I, and ECL_{III} that critical for recognition and binding of ziconotide in $\text{Ca}_v2.2$ are not conserved in the $\text{Ca}_v2.3$. We speculate these discrepancies may also contribute to insensitivity of $\text{Ca}_v2.3$ to ziconotide. We included the structural comparison in the newly added Supplementary Figure 3 and added a brief discussion about this.

In the revised line 142, it reads “Moreover, other residues on P-loops and ECLs that are involved in ziconotide binding are also not conserved in $\text{Ca}_v2.3$ (Supplementary Figure 3), rendering $\text{Ca}_v2.3$ insensitive to the ziconotide.”.

We also attached the Supplementary Figure 3 here for your convenience.

a. Putative structural mismatches between the Cav_v2.3 and ziconotide. Extracellular helices (ECLs) and the P-loops (P1 and P2) of the Cav_v2.3 (blue) are superimposed on the corresponding structures of the ziconotide-bound Cav_v2.2. The ziconotide is shown as transparent green surface. Non-conserved residues that contribute to the structural mismatch between Cav_v2.3 and ziconotide are shown as sticks, overlaid by red surfaces, and indicated in **(b)**. **b.** Sequences alignments between Cav_v2.3 and Cav_v2.2 at the P-loops and ECLs. Conserved residues are highlighted in blue. Non-conserved residues that form close contact with ziconotide in Cav_v2.2 are indicated using red triangles.

Further structural analysis revealed that the Q939 located on S5_{III} in Cav_v1.1 (Q1010 in Cav_v1.2) is substituted by M1300 in the Cav_v2.3. A previous study did show that the Q1010M mutation abolished the dihydropyridine sensitivity of Cav_v1.2^[1]. We have also discussed this site and added references in the revised manuscript. In the revised line 135, it now reads: “Meanwhile, a previous study reported that the Q1010 of Cav_v1.2 is important for sensitivity to dihydropyridine (DHP) and the Q1010M mutant had a decreased sensitivity to DHP molecules³⁵. The equivalent position in Cav_v2.3 is occupied by M1300, thus also contributing to the pharmacoresistance of Cav_v2.3 to DHP molecules.”.

The M1300 (Cav_v2.3) and the corresponding Q939 (Cav_v1.1) have been shown as sticks in the revised Figure 1d, which is attached below for your convenience.

References:

[1] He, Ming, et al. "Motif III S5 of L-type calcium channels is involved in the dihydropyridine binding site: a combined
radioligand binding and electrophysiological study." *Journal of Biological Chemistry* 272.5 (1997): 2629-2633.

*4. Figure 1C:*

*The pore profiling shows that they determined a closed state of the calcium channel. Instead of presenting 1.0-Å pore*
*radius, I believe that presenting a more relevant pore radius such as a calcium ion radius (dehydrated and hydrated)*
*can improve this panel.*

**Reply:** We thank the reviewer for this comment. The 1.0-Å pore radius criterion is based on the ionic radius of calcium,
which comes about at 1.00 Å^[1-2]. We have clarified this in the revised figure legend. It now reads: "...The vertical dashed
line marks the 1.0-Å pore radius, which represents the ionic radius of calcium."

References:

[1] Shannon, Robert D. "Revised effective ionic radii and systematic studies of interatomic distances in halides and
chalcogenides." *Acta crystallographica section A: crystal physics, diffraction, theoretical and general crystallography*
32.5 (1976): 751-767.

[2] ChemGlobe database (https://chemglobe.org/ptoe/_/20.php)

*5. Supplementary Figure 3.*

*This figure needs improved labeling to give more information to readers. This figure presents evidence to support the*
*authors' statements in lines 142 - 174, thus should corresponding labelings should be presented here. For example, the*
*addition of 'S4-S5' labeling will help readers in understanding what they are saying in lines 150-152.*

**Reply:** We thank the reviewer for this comment. More labels have been added in the revised Supplementary Figure 4
to better illustrate the structure comparisons. The revised Supplementary Figure 4b-4d is attached below for your
convenience.

i4 *6. EM validation report*

i5 *Magnification information in the EM validation report is different from Supplementary Table 1 and the method part.*
i6 *Please correct it.*

i7 **Reply:** We thank the reviewer for pointing out this. The actual nominal magnification should be $\times 130,000$ as was
i8 reported in the Method section and Supplementary Table 1. This value had been updated in the wwPDB deposition.

Reviewers' Comments:

Reviewer #1:

Remarks to the Author:

The authors have made all the corrections that I requested.

The manuscript can be accepted for publication after correcting one minor remark.

line 117. Please indicate where the ECL is located between S5 and P1 or between P2 and S6.

Reviewer #2:

Remarks to the Author:

I have reviewed a revised version of the manuscript entitled Molecular insights into the gating mechanisms of voltage-gated calcium channel CaV2.3, by Y. Gao et al.

The authors have constructively responded to all my comments and modified the manuscript accordingly. However, I suggest that the Author briefly discuss their results in view of the recent work by Xia Yao et al [1] (see reference below) particularly regarding PIP2 which was found to bind to the interface of VSDII and the pore domain.

1) Yao, X., Wang, Y., Wang, Z. et al. Structures of the R-type human Cav2.3 channel reveal conformational crosstalk of the intracellular segments. Nat Commun 13, 7358 (2022)

<https://doi.org/10.1038/s41467-022-35026>

I have no further questions.

Reviewer #3:

Remarks to the Author:

All my concerns have been properly addressed in the revisions.

I would ask about one minor point as below.

The running buffer condition in line 423 is, "20 mM HEPES pH 7.5, 1.5 mM NaCl, 5 mM β -ME, and 0.01 % (w/v) GDN".

I would ask double check the condition. 1.5 mM NaCl concentration looks much lower than usual, so I think it is a typo.

Point-by-point response for

Molecular insights into the gating mechanisms of voltage-gated calcium channel Cav2.3

**TO REVIEWER REPORT #1**

*The authors have made all the corrections that I requested.*

*The manuscript can be accepted for publication after correcting one minor remark.*

*line 117. Please indicate where the ECL is located between S5 and P1 or between P2 and S6.*

**Reply:** We are deeply indebted to the reviewer for his/her comments that greatly improved this study.

In the revised line 114, it now reads: "Similar to other Cav channels, Cav2.3 harbors four extracellular loops (ECLs) that
0 are also positioned between S5 and P1, as well as P2 and S6 helices in the pore domain (Figure 1a–1c).".

**TO REVIEWER REPORT #2**

*I have reviewed a revised version of the manuscript entitled Molecular insights into the gating mechanisms of voltage-*
*gated calcium channel Cav2.3, by Y. Gao et al.*

*The authors have constructively responded to all my comments and modified the manuscript accordingly. However, I*
*suggest that the Author briefly discuss their results in view of the recent work by Xia Yao et al [1] (see reference below)*
*particularly regarding PIP2 which was found to bind to the interface of VSDII and the pore domain.*

*1) Yao, X., Wang, Y., Wang, Z. et al. Structures of the R-type human Cav2.3 channel reveal conformational crosstalk of*
*the intracellular segments. Nat Commun 13, 7358 (2022) <https://doi.org/10.1038/s41467-022-35026>*

*I have no further questions.*

**Reply:** We are grateful for the reviewer's comments that helped us to revise the manuscript.

In the revised line 190, it now reads: "However, another recent structural investigation of Cav2.3 channel showed that
PIP₂ could bind to this site but is not responsible to the resting state of VSDII³⁹."

**TO REVIEWER REPORT #3**

*All my concerns have been properly addressed in the revisions.*

*I would ask about one minor point as below.*

*The running buffer condition in line 423 is, "20 mM HEPES pH 7.5, 1.5 mM NaCl, 5 mM β-ME, and 0.01 % (w/v) GDN".*

*I would ask double check the condition. 1.5 mM NaCl concentration looks much lower than usual, so I think it is a typo.*

**Reply:** We greatly appreciate the reviewer's comment and have corrected this typo in the revised submission. In the
revised line 417, it now reads: "... a running buffer containing 20 mM HEPES pH 7.5, 150 mM NaCl, 5 mM β-ME, ...".